# More accurate quantification of model-to-model agreement in externally forced climatic responses over the coming century

Nicola Maher [1✉], Scott B. Power [2,3,4] & Jochem Marotzke [1]

Separating how model-to-model differences in the forced response ($U_{MD}$) and internal variability ($U_{IV}$) contribute to the uncertainty in climate projections is important, but challenging. Reducing $U_{MD}$ increases confidence in projections, while $U_{IV}$ characterises the range of possible futures that might occur purely by chance. Separating these uncertainties is limited in traditional multi-model ensembles because most models have only a small number of realisations; furthermore, some models are not independent. Here, we use six largely independent single model initial-condition large ensembles to separate the contributions of $U_{MD}$ and $U_{IV}$ in projecting 21st-century changes of temperature, precipitation, and their temporal variability under strong forcing (RCP8.5). We provide a method that produces similar results using traditional multi-model archives. While $U_{MD}$ is larger than $U_{IV}$ for both temperature and precipitation changes, $U_{IV}$ is larger than $U_{MD}$ for the changes in temporal variability of both temperature and precipitation, between 20° and 80° latitude in both hemispheres. Over large regions and for all variables considered here except temporal temperature variability, models agree on the sign of the forced response whereas they disagree widely on the magnitude. Our separation method can readily be extended to other climate variables.

[1] Max Planck Institute for Meteorology, Hamburg, Germany. [2] School of Earth, Atmosphere and Environment, Monash University, Melbourne, Victoria, Australia. [3] ARC Centre of Excellence for Climate Extremes, Monash University, Melbourne, Victoria, Australia. [4] Bureau of Meteorology, Docklands, Victoria, Australia. ✉email: nicola.maher@colorado.edu

dentifying whether different climate models agree in their response to external forcing is important for assessing confidence in future projections[1]. However, in practice, climate projections are often made using a multi-model ensemble[2,3], where assessing this agreement can be difficult. Part of the problem arises because under the same forcing scenario, projections from two different model runs can differ because of internal variability, as well as differences in the forced response to external forcing in the models. The other part of the problem arises because climate models are not all independent, with many models sharing components and code[1,4–6]. This reduces the effective number of degrees of freedom of an ensemble containing such models, which can in turn lead to both an overestimate of the statistical significance of projected changes and overconfidence[1,7]. Here, we quantify how model-to-model differences and internal variability contribute to the uncertainty in projections of temperature and precipitation and their temporal variability, without relying on the assumptions of previous methods.

The contribution of internal variability to the uncertainties in projections tends to be larger on short timescales and smaller spatial scales, while model-to-model differences in the response to external forcing play a larger role on longer timescales and over larger domains[8–13]. In addition, the relative roles of these two uncertainties depend on the quantity considered[9,10,13] and the location investigated[1], making it non-trivial to draw generalisations about which uncertainty dominates. Quantifying the role of internal variability in a multi-model ensemble such as CMIP[2,3] is difficult due to the limited number of simulations available for each model. Partitioning has previously been achieved by using statistical methods to estimate the forced signal[9,10] or by using a long pre-industrial control to estimate internal variability[14,15]. However, these methods are limited as they make some ad hoc assumptions in their estimates of the forced response and cannot account for changes in internal variability itself[13,16]. Indeed, Lehner et al.[13] recently demonstrated that the regional errors in the partitioning of uncertainty can be as large as 50% using the traditional approaches.

Methods to account for the lack of independence of the climate models include institutional democracy[7], the ensemble-mean performance[17] and multi-family ensembles[1]; these methods are used to weight models when making projections. Other methods determine weights using a combination of model performance in comparison to observations and the similarity of each model's response[18–20]. These methods are, however, not immune to the role of internal variability[21], and the same model can have substantially different weights, depending on which ensemble member is used[22]. In general, community consensus appears to be that there is no one-size-fits-all approach to model independence and performance weighting, and that the method used must be application dependent[5].

Confidence in model projections has been characterised using a variety of methods. Some demonstrate the percentage of models that agree on the sign of the change[1,23], with others additionally including the agreement on no change[1,24]. Some investigate the signal-to-noise ratio[25], while others use statistical significance levels under the assumption of model independence[1]. When the last Intergovernmental Panel on Climate Change (IPCC) report investigated these different methods, they found that their assessment of confidence did not agree[23]. When considering temperature, precipitation and their temporal variability, studies have usually used the percentage of models that agree on the sign of the change to assess confidence, with different thresholds used in different studies, ranging from 67 to 90%[26–29]. While exceptions exist[1], model independence is not usually included in these estimates of confidence, although some studies exclude models that are shown to perform poorly[30].

In many previous projections, including most of the last IPCC assessment, only one ensemble member per model is used (r1i1p1)[23]. The choice to use a single member, to use all members or to take the ensemble mean of those models that have more than one member can be somewhat arbitrary, with no clear consensus on which method is best nor on how this affects confidence. As such, currently, there is no best practice for how to deal with such a multi-model ensemble[5]. More recently, Merrifield et al.[22] proposed a weighting scheme that can deal with a multi-model ensemble that includes many ensemble members from some models. This provides a new opportunity to include all members of a multi-model ensemble in the estimate; however, this, and previous model weighting methods can also only be applied to the multi-model ensemble mean, and cannot be used to assess the relative roles of model-to-model differences and internal variability in causing the ensemble spread.

Model-to-model differences and internal variability can now be better quantified using a single model initial-condition large ensembles (SMILEs)[12,13]. SMILEs are based on individual climate models that are run many times from differing initial conditions[16,31–33]. At any point in time, the range of a quantity in each SMILE can be used to quantify the model's internal variability, while the mean across the ensemble provides an unbiased estimate of each individual SMILE's response to external forcing[16]. While single SMILEs have now been used in many studies[31], a new archive of SMILEs will allow comparisons across multiple models[33]. Importantly, this archive can be used to assess the confidence in projections under increasing greenhouse gas emissions, which is particularly important for variables such as temperature and precipitation, due to their potential impacts on people and ecosystems[34].

In this study, we use six SMILEs to show that under strong forcing, model-to-model differences between simulated twenty-first-century changes are almost always larger than the internal variability of temperature and precipitation. For temporal temperature and precipitation variability, the internal variability in the projections is larger than the model-to-model differences in the extratropics, with model-to-model differences either similar in magnitude or larger than the internal variability elsewhere on the globe. We show that the sign of the projected change agrees for much of the globe, while models disagree on the magnitude of the projected change.

## Results

**Separating the forced response and internal variability.** Using six SMILEs, we estimate the contribution of model-to-model differences and internal variability in causing uncertainty in the projections of future climate under strong forcing. We calculate the projected change in an individual ensemble member: the forced response both in each individual SMILE and across the six SMILEs, which is an estimate of the response due to external forcing alone, and the uncertainty in projections due to model-to-model differences in the forced response ($U_{MD}$) and internal variability ($U_{IV}$). These quantities can be described for temperature ($T$) and using the following equations:

The projected change in $T$ in a single ensemble member (e) of a single SMILE (s) is

$$\Delta T_{s,e} = (\bar{T}_{s,e,21C} - \bar{T}_{s,e,20C}) \tag{1}$$

where $T_{s,e}$ is temperature from a single ensemble member and the overbar indicates a time average over 2050–2099 from RCP8.5 (21C) and over 1950–1999 from the historical simulations (20C).

The forced response in $T$ in a single SMILE (s) is calculated as the ensemble mean of the projected change

$$\Delta T_{s,\mathrm{F}} = \frac{1}{e_s} \sum_{e=1}^{e_s} \Delta T_{s,e} \qquad (2)$$

where $e_s$ is the ensemble size for each individual SMILE.

The multi-ensemble-mean forced response in $T$ for the six SMILEs is the mean across the six individual SMILE ensemble means

$$\Delta T_{\mathrm{F}} = \frac{1}{n} \sum_{s=1}^{n} \Delta T_{s,\mathrm{F}} \qquad (3)$$

where $n$ is the number of SMILEs.

The spread in $\Delta T$ across a SMILE (s) due to internal variability is calculated as the sample standard deviation of the projected change across the ensemble members of the SMILE:

$$\sigma(\Delta T_s) = \sqrt{\frac{1}{e_s - 1} \sum_{e=1}^{e_s} (\Delta T_{s,e} - \Delta T_{s,\mathrm{F}})^2} \qquad (4)$$

An estimate of the uncertainty in $\Delta T$ due to internal variability can be expressed as an average of the internal variability across the six SMILEs:

$$U_{\mathrm{IV}} = \sqrt{\frac{1}{n} \sum_{s=1}^{n} \sigma^2(\Delta T_s)} \qquad (5)$$

The variance of the forced response is estimated using

$$\sigma^2_{\mathrm{FR}} = D^2 - E^2 \qquad (6)$$

where $D^2$ is the sample variance of the ensemble means calculated as follows:

$$D^2 = \frac{1}{n-1} \sum_{s=1}^{n} (\Delta T_{s,\mathrm{F}} - \Delta T_{\mathrm{F}})^2 \qquad (7)$$

and $E^2$ is included to offset the contribution of internal variability to the variance of the ensemble means (see Rowell et al.[35] for further details) and is equal to the average value of $\sigma^2(\Delta T_s)/e_s$ across the six ensembles. This term is discussed in more detail in "Methods". This offset is expressed as

$$E^2 = \frac{1}{n} \sum_{s=1}^{n} \frac{\sigma^2(\Delta T_s)}{e_s} \qquad (8)$$

The uncertainty in $\Delta T$ due to model differences can be quantified as the square root of the variance of the forced response:

$$U_{\mathrm{MD}} = \sqrt{\sigma^2_{\mathrm{FR}}} \qquad (9)$$

In this study, we will investigate the externally forced response of annual-mean temperature ($T$), annual-mean precipitation ($P$), annual-mean temporal temperature variability ($T_\sigma$) and annual-mean temporal precipitation variability ($P_\sigma$). The equations for $P$ can be found by replacing $T$ with $P$ in Eqs (1)–(9). To compute $T_\sigma$ first, we remove the forced response in $T$ by removing the ensemble mean at each timestep. We then calculate $T_\sigma$ in each ensemble member as the sample standard deviation over the time period 2050–2099 from RCP8.5 and the sample standard deviation over the time period 1950–1999 from the historical simulations.

The projected change in $T_\sigma$ in a single ensemble member (e) of a single SMILE (s) is

$$\Delta T_{\sigma,s,e} = (T_{\sigma,s,e,21\mathrm{C}} - T_{\sigma,s,e,20\mathrm{C}}) \qquad (10)$$

where $T_{\sigma,s,e,21\mathrm{C}}$ indicates the time period 2050–2099 (21C) and $T_{\sigma,s,e,20\mathrm{C}}$ is from the period 1950–1999 (20C).

The forced response in $T_\sigma$ in a single SMILE (s) is

$$\Delta T_{\sigma,s,\mathrm{F}} = \sqrt{\frac{1}{e_s} \sum_{e=1}^{e_s} (T_{\sigma,s,e,21\mathrm{C}})^2} - \sqrt{\frac{1}{e_s} \sum_{e=1}^{e_s} (T_{\sigma,s,e,20\mathrm{C}})^2} \qquad (11)$$

Here, the standard deviation is calculated individually for each time period as the square root of the ensemble-mean variance before the difference between the two time periods is calculated. The multi-ensemble mean forced response in $T_\sigma$ for the six SMILEs is

$$\Delta T_{\sigma,\mathrm{F}} = \sqrt{\frac{1}{n} \sum_{s=1}^{n} \left[ \frac{1}{e_s} \sum_{e=1}^{e_s} (T_{\sigma,s,e,21\mathrm{C}})^2 \right]} - \sqrt{\frac{1}{n} \sum_{s=1}^{n} \left[ \frac{1}{e_s} \sum_{e=1}^{e_s} (T_{\sigma,s,e,20\mathrm{C}})^2 \right]}$$

$$(12)$$

Here, the individual standard deviation is calculated for each time period as the square root of the multi-ensemble-mean variance of the six SMILEs. The equations for $\sigma(\Delta T_{\sigma,s})$, and $U_{\mathrm{IV}}$ and $U_{\mathrm{MD}}$ for $T_\sigma$ can be found by replacing $\Delta T$ with $\Delta T_\sigma$ in Eqs. (4)–(9). The equations for $P_\sigma$ can be found by replacing $T_\sigma$ in Eqs. (10)–(12), and $\Delta T$ in Eqs. (4)–(9) with $\Delta P_\sigma$. Discussion on Eqs. (10)–(12) can be found in "Methods".

We utilise a new archive of SMILEs (Supplementary Table 1; Deser et al.[33]), which consists of the following six models: CanESM2[32], CESM-LE[31], CSIRO-Mk3-6-0[36], GFDL-CM3[37], GFDL-ESM2M[38] and MPI-GE[16]. The six SMILEs have different numbers of ensemble members. The smallest ensemble has 20 members, while the largest has 100. The estimate of the internal variability of the forced response obtained from each SMILE is found to be a model quantity that is not related to the ensemble size, i.e., having a larger ensemble does not increase the magnitude of the estimate of internal variability (see Supplementary Note 1 for discussion and Supplementary Figs. 1–4). We find, however, that smaller ensemble sizes do result in greater uncertainty in the magnitudes of both the forced response and internal variability of the forced response in our analysis. The ensemble size needed is larger for changes in temporal variability than mean-state changes in agreement with previous work[39,40]. An in-depth analysis of the uncertainties caused by the varying ensemble sizes of the SMILEs used in this study can be found in Supplementary Note 1.

For the methods used in this study to perform best, we would ideally use a set of SMILEs that are independent and cover the current range of global coupled climate models available. The SMILEs used in this study were picked due to their availability and their minimum ensemble size of 20[33] and because two previous studies showed that they cover the range of the CMIP5 models well[12,13]. While it can be difficult to assess whether models share pieces of code, it is possible to assess whether they share components, such as the ocean or atmosphere[6]. Although two of the SMILEs (GFDL-ESM2M and GFDL-CM3) share the ocean, sea-ice and land components, and have a similar atmospheric model, and a third (CSIRO-Mk3-6-0) uses an older version of the same ocean when assessing precipitation biases over Southern Asia, it has been shown that all three of these models behave independently[41]. Differences between GFDL-ESM2M and GFDL-CM3 are also investigated by Lehner et al.[13]. They find that the models behave differently for global and British Isles annual decadal mean temperature and global annual, Sahel summer and Southern European Summer decadal mean precipitation, although they behave more similarly in the Southern Ocean for decadal annual-mean temperature. The rest of the SMILEs do not share any components[6]. Here, given the two SMILEs that share components have been shown to behave differently for a range of variables, we treat each SMILE as

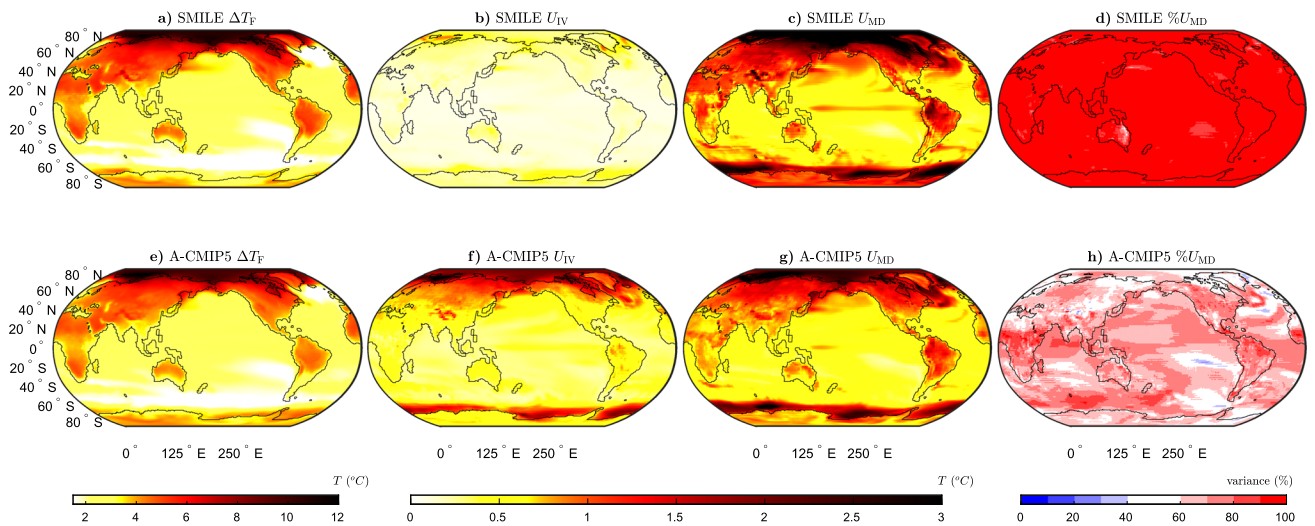

**Fig. 1 Long-term projections of the mean-state temperature response to external forcing (ΔT) and the associated uncertainties. a–d** The six single model initial-condition large ensembles (SMILEs), and **e, f** CMIP5 multi-model ensemble using the atmospheric sub-ensemble method (A-CMIP5; see "Methods" for details) for the period 2050–2099 (RCP8.5 forcing) as compared to 1950–1999 (historical forcing). **a, e** Multi-ensemble-mean forced response ($\Delta T_F$), **b, f** uncertainty due to internal variability ($U_{IV}$), **c, g** uncertainty due to model-to-model differences ($U_{MD}$) and **d, h** percentage variance contribution of $U_{MD}$ to the sum of $U_{MD}$ and $U_{IV}$.

independent so as not to decrease the number of SMILEs; this independence assumption is validated later in the study where the three aforementioned models do not cluster together for the quantities considered.

While uncertainties in the estimation of the forced response and internal variability exist due to the varying ensemble sizes and the availability of models in the large-ensemble archive, the use of SMILEs allows us to simply estimate both the internal variability and the forced response more accurately than can be done with single runs or much smaller ensembles. This means that $U_{MD}$ and $U_{IV}$ can be easily calculated. Importantly, using SMILEs has the advantage that we can more accurately assess the role of each uncertainty in projections of temperature and precipitation temporal variability themselves. $\Delta_{s,F}$ and $\sigma(\Delta_s)$ are shown for each individual SMILE in Supplementary Figs. 5–8.

**Mean-state projections in the SMILEs.** We first use the six SMILEs to illustrate the forced response in temperature ($\Delta T_F$, Fig. 1a). We find that the land surface is projected to warm more than the ocean, the Arctic has the largest projected temperature increases and the areas with the smallest warming are the Southern Ocean and the North Atlantic warming hole in agreement with the previous work[23,29,42,43]. By using SMILEs, we are able to precisely quantify the magnitude of $U_{IV}$ and $U_{MD}$ (Fig. 1b, c). In general, $U_{MD}$ is the largest over land, the high-latitude oceans and the tropical Pacific. The largest magnitudes of $U_{MD}$ are found in the Arctic and over the Southern Ocean. The magnitude of $U_{IV}$ does not vary much across the globe. We can also assess the importance of $U_{MD}$ by comparing $U_{MD}$ and $U_{IV}$ and computing the percentage of the combined variance of the two quantities due to $U_{MD}$ (Fig. 1d). Where $U_{IV}$ is of a similar magnitude to $U_{MD}$, an individual SMILE could cover the uncertainty in $U_{MD}$ itself. In these regions, even if the models agree on the projected change, the range of changes that could be observed is considerable due to the large $U_{IV}$. Conversely, $U_{MD}$ is most important where these model-to-model differences are much larger than $U_{IV}$ (Fig. 1d, red regions). We find that for $\Delta T$, $U_{MD}$ is larger than $U_{IV}$ in almost all areas of the globe, except the Eastern Australian coastline where the two contributions are of similar magnitude.

We next consider precipitation (Fig. 2). We find, similar to previous work[23,26], that $\Delta P_F$ shows a large increase in the tropical Pacific, a decrease over North-Eastern South America and Southern Africa, a decrease over most of the subtropical Southern Hemisphere and an increase over most of the high latitudes in both hemispheres. We again precisely quantify the uncertainties and find that for $\Delta P$, $U_{MD}$ is the largest between approximately 20ºS and 20º N, while $U_{IV}$ is the largest in the Western tropical Pacific, and very small in the high latitudes. Elsewhere across the globe, $U_{IV}$ is generally homogeneous in magnitude. We find that $U_{MD}$ is larger than $U_{IV}$ in most regions of the globe, with small areas where the two uncertainties are of similar magnitude or $U_{IV}$ is larger. Overall, for both long-term mean-state projections of $\Delta T$ and $\Delta P$, $U_{MD}$ dominates across most of the globe. These results confirm the general results from previous studies, which have shown that $U_{MD}$ is much more important than $U_{IV}$ on longer timescales for both $\Delta T$ and $\Delta P$[9,10,13]. This additionally tells us that by understanding why the externally forced responses differ and improving model-to-model agreement in those same responses in the future, we can reduce the uncertainty in long-term projections for these variables.

**Variability projections in the SMILEs.** While previous studies have used estimates to partition uncertainty into $U_{MD}$ and $U_{IV}$ for temperature and precipitation projections[9,10,13], they have been not been able to quantify these uncertainties in the temporal variability. In this section, we fill this gap by partitioning the uncertainties as in Figs. 1 and 2 for $\Delta T_\sigma$ and $\Delta P_\sigma$. The forced response in temporal temperature variability itself ($\Delta T_{\sigma_F}$, Fig. 3a) is qualitatively in agreement with previous work[27,29,44,45], showing a general increase over the Southern Hemisphere land masses and Africa, an increase over the Northern Hemisphere subtropical land surface and a decrease over the Northern Hemisphere high latitudes and the Southern Ocean. We find that $U_{IV}$ for $\Delta T_\sigma$ is the largest over the Arctic, the Northern Hemisphere high-latitude land surface the Southern Ocean near the continent edges, parts of Australia and the tropical Pacific (Fig. 3b). $U_{MD}$ is the largest over the high-latitude oceans, the tropical Pacific and in patches over the land surface. When considering the relative magnitudes of the uncertainties (Fig. 3d),

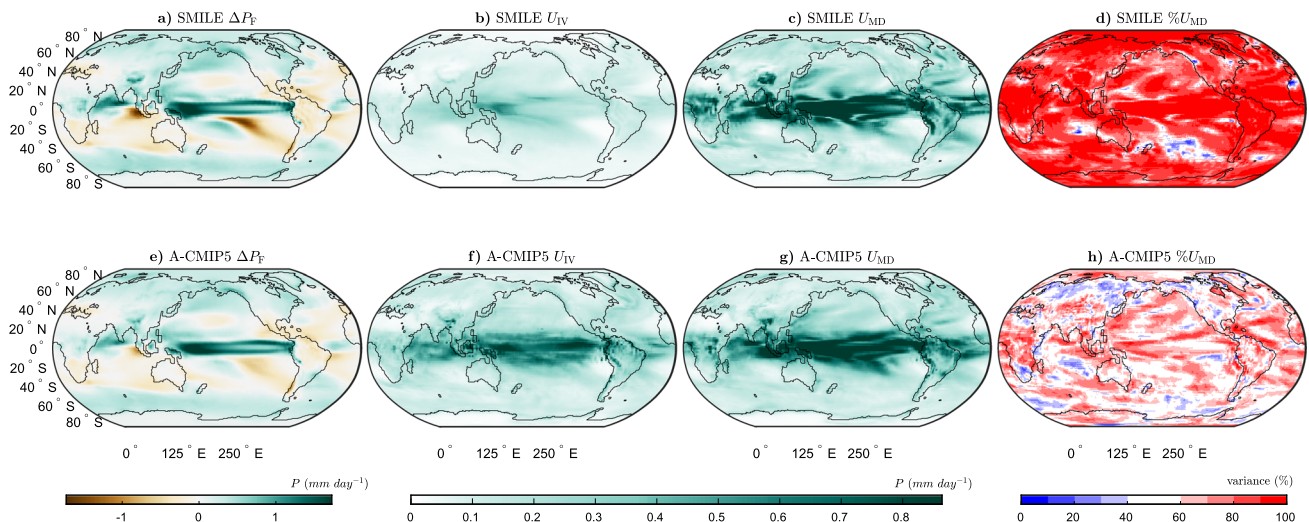

**Fig. 2 Long-term projections of the mean-state precipitation response to external forcing (ΔP) and the associated uncertainties. a–d** The six single model initial-condition large ensembles (SMILEs), and **e**, **f** CMIP5 multi-model ensemble using the atmospheric sub-ensemble method (A-CMIP5, see "Methods" for details) for the period 2050–2099 (RCP8.5 forcing) as compared to 1950–1999 (historical forcing). **a**, **e** Multi-ensemble-mean forced response ($\Delta P_F$), **b**, **f** Uncertainty due to internal variability ($U_{IV}$), **c**, **g** uncertainty due to model-to-model differences ($U_{MD}$) and **d**, **h** percentage variance contribution of $U_{MD}$ to the sum of $U_{MD}$ and $U_{IV}$.

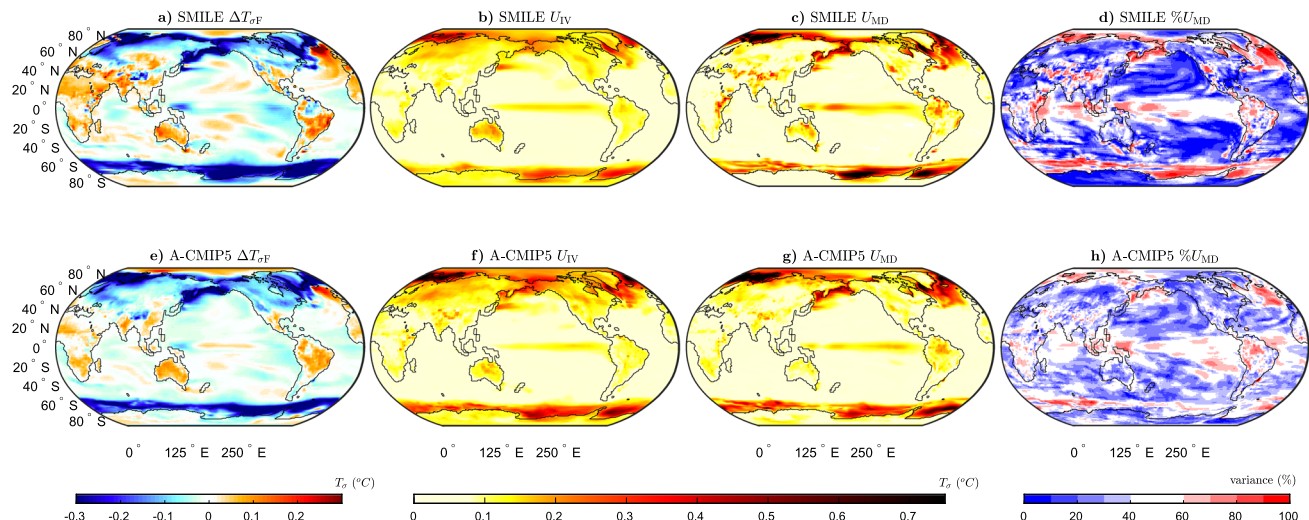

**Fig. 3 Long-term projections of the temporal temperature variability response to external forcing ($\Delta T_\sigma$) and the associated uncertainties. a–d** The six single model initial-condition large ensembles (SMILEs), and **e**, **f** CMIP5 multi-model ensemble using the atmospheric sub-ensemble method (A-CMIP5, see Methods for details) for the period 2050–2099 (RCP8.5 forcing) as compared to 1950–1999 (historical forcing). **a**, **e** Multi-ensemble-mean forced response ($\Delta T_{\sigma_F}$), **b**, **f** uncertainty due to internal variability ($U_{IV}$), **c**, **g** uncertainty due to model-to-model differences ($U_{MD}$) and **d**, **h** percentage variance contribution of $U_{MD}$ to the sum of $U_{MD}$ and $U_{IV}$.

$U_{IV}$ is often larger or the same magnitude as $U_{MD}$. In the tropics, the two uncertainties are of similar magnitude or $U_{MD}$ is larger. Elsewhere $U_{IV}$ is larger, except in the Southern Ocean near the Antarctic continent and in patches of the Northern Hemisphere high-latitude oceans, where $U_{MD}$ is larger than $U_{IV}$.

We next show the same breakdown for $\Delta P_\sigma$ in Fig. 4. Similar to previous work, $\Delta P_{\sigma_F}$ increases in the tropical Pacific and generally over the globe, with regions of decreasing variability over the subtropical South-Eastern Pacific Ocean, the subtropical Atlantic ocean and off the coast of South-Western Australia[26]. Unlike Pendergrass et al.[26], we find a decrease in $\Delta P_{\sigma_F}$ over the subtropical North-Eastern Pacific, as well as over Northern South America. These differences presumably occur because we consider annual variability, while Pendergrass et al.[26] consider

seasonal and daily variability. Both $U_{IV}$ and $U_{MD}$ are the largest in the tropical Pacific, extending into the far Eastern Indian Ocean and over the Indonesian region. For $\Delta P_\sigma$, $U_{IV}$ is more important than $U_{MD}$ poleward 20° in both hemispheres. In the tropics, $U_{MD}$ is generally larger than $U_{IV}$. Unlike for $\Delta T$ and $\Delta P$, we have shown that for temporal variability projections over long timescales, $U_{MD}$ does not necessarily dominate over $U_{IV}$. For $\Delta T_\sigma$, $U_{IV}$ is generally the same magnitude or larger than $U_{MD}$, except in parts of the tropics, the Southern Ocean and parts of the far Northern Hemisphere oceans, while for $\Delta P_\sigma$, $U_{IV}$ is larger than $U_{MD}$ for most regions outside the tropics. This result has implications for understanding the spread of projected changes, because in these regions, improving model-to-model agreement may only have a limited impact on the spread of projections.

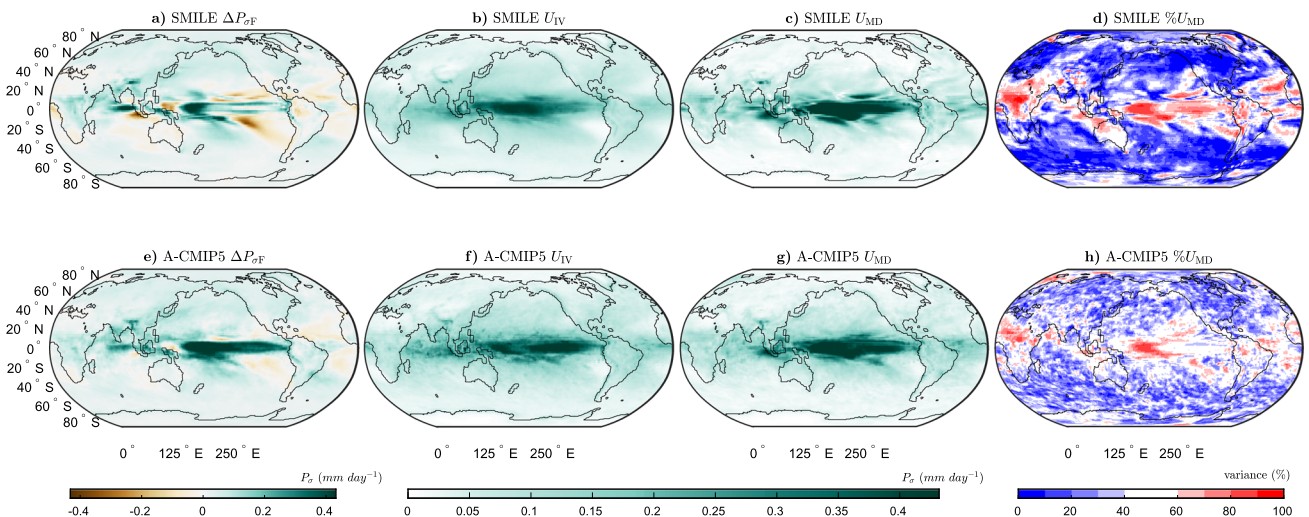

**Fig. 4 Long-term projections of the temporal precipitation variability response to external forcing ($\Delta P_\sigma$) and the associated uncertainties. a–d** The six single model initial-condition large ensembles (SMILEs), and **e, f** CMIP5 multi-model ensemble using the atmospheric sub-ensemble method (A-CMIP5, see "Methods" for details) for the period 2050–2099 (RCP8.5 forcing) as compared to 1950–1999 (historical forcing). **a, e** Multi-ensemble-mean forced response ($\Delta P_{\sigma_F}$), **b, f** uncertainty due to internal variability ($U_{IV}$), **c, g** uncertainty due to model-to-model differences ($U_{MD}$) and **d, h** percentage variance contribution of $U_{MD}$ to the sum of $U_{MD}$ and $U_{IV}$.

**A new methodology for multi-model ensembles**. The SMILE analysis to separate $U_{IV}$ and $U_{MD}$ was not previously possible with an ensemble of opportunity, such as CMIP5. We overcome this limitation by treating models with the same atmospheric component as the same "family" (see "Methods", Supplementary Tables 3, and ref. [6]). The analogous approach is used for models sharing the ocean component (Supplementary Table 4 and Supplementary Fig. 9). By creating these sub-ensembles, we can now estimate the forced response and internal variability for sets of models that share components. We note that the sub-ensembles include both different models, which share an atmospheric component, and multiple ensemble members from the same model where available. The general features and patterns found from the SMILE analysis are remarkably well captured using this methodology (Figs. 1–4), demonstrating that this method can be used to estimate $U_{IV}$ and $U_{MD}$, without the need for many large ensembles.

Differences between the SMILE analysis and the CMIP5 atmospheric sub-ensemble method used in this study could occur for two main reasons. First, $U_{IV}$ could be overestimated in the CMIP5 analysis as the models in the sub-ensembles are not the same. This means that the uncertainty estimated as $U_{IV}$ contains some of the uncertainty from $U_{MD}$. Second, $U_{MD}$ could be underestimated in both analyses. This could occur in the CMIP5 analysis because some of $U_{MD}$ is included in the estimate of $U_{IV}$. It could also occur in the SMILE analysis as all models in CMIP5 are not available in the SMILE archive.

We find that $\Delta T_F$ is larger in the SMILEs than in CMIP5, except for in parts of the Southern Ocean, the North Atlantic and in the Arctic above Europe (Supplementary Fig. 10a). This could occur due to the different models used in the different analyses or due to sampling errors in CMIP5. We find that for the mean-state projections, $U_{IV}$ is generally smaller and $U_{MD}$ is larger in the SMILE analysis (Supplementary Fig. 11a, b, e, f), likely due to the overestimation of internal variability in the CMIP5 analysis. We find that the estimate of $U_{IV}$ for $\Delta T$ is much larger, and $U_{MD}$ is slightly smaller when using CMIP5 compared to the SMILEs (Supplementary Fig. 11a, e). For $\Delta P$, the two methods show differences mainly between 20ºN and 20ºS (Supplementary Fig. 10e–g). $U_{IV}$ is again overestimated by the CMIP5 analysis (Supplementary Fig. 11b), although the magnitude of this

overestimation is much less than for $\Delta T$. When considering $U_{IV}$ for $\Delta T_\sigma$, the differences between the two methods are small, except in the far high-latitude oceans where $U_{IV}$ is somewhat overestimated by the CMIP5 sub-ensembles (Supplementary Figs. 10j and 11c). $U_{MD}$ for $\Delta T_\sigma$ shows regions of both over- and underestimation; however, this quantity is more likely to be overestimated globally (Supplementary Figs. 10k and 11g). For $\Delta P_\sigma$, the main differences between the methods are found in the tropical Pacific (Supplementary Fig. 10m–o). For both $\Delta T_\sigma$ and $\Delta P_\sigma$, the ratio of $U_{IV}$ between the two methods is much closer to one than for $\Delta T$ and $\Delta P$ (Supplementary Fig. 11c, d).

For the four quantities considered ($\Delta T$, $\Delta P$, $\Delta T_\sigma$ and $\Delta P_\sigma$), the two methods generally agree on whether $U_{IV}$ or $U_{MD}$ is larger (Supplementary Fig. 10d, h, l, p). This increases confidence in our assessment of the relative importance of each uncertainty. It also indicates that the methodology using the sub-ensembles can generally provide a reasonable assessment of the relative importance of $U_{MD}$ and $U_{IV}$ to the projection uncertainty when only a single member or small ensemble is available for some of the models. This should prove useful for understanding uncertainty in the CMIP6 models database that is currently being developed if, as expected, many modelling groups only provide a single or a small number of ensemble members for a given forcing scenario.

**Global assessment of model-to-model agreement**. In the previous sections, we quantified $U_{MD}$ and $U_{IV}$ and compared their relative magnitudes. Now, we investigate model-to-model agreement on the sign of the forced response in 2050–2099 as compared to 1950–1999 for the same four quantities ($\Delta T$, $\Delta P$, $\Delta T_\sigma$ and $\Delta P_\sigma$; Fig. 5). We show SMILE agreement on the sign of the change across the globe in colour (red for an increase and blue for a decrease) and the CMIP5 agreement, using the atmospheric sub-ensemble method, in stippling (dots for a decrease and plus signs for an increase). $\Delta T_F$ increases at all locations in all six SMILEs, except the North Atlantic Ocean and two patches in the Southern Ocean. For CMIP5, we find agreement on an increase in $\Delta T_F$ at all locations. Model-to-model agreement in $\Delta T_{\sigma_F}$ is more fragmented. Small areas in the tropics and extratropics show model-to-model agreement, such as the far-East Pacific where

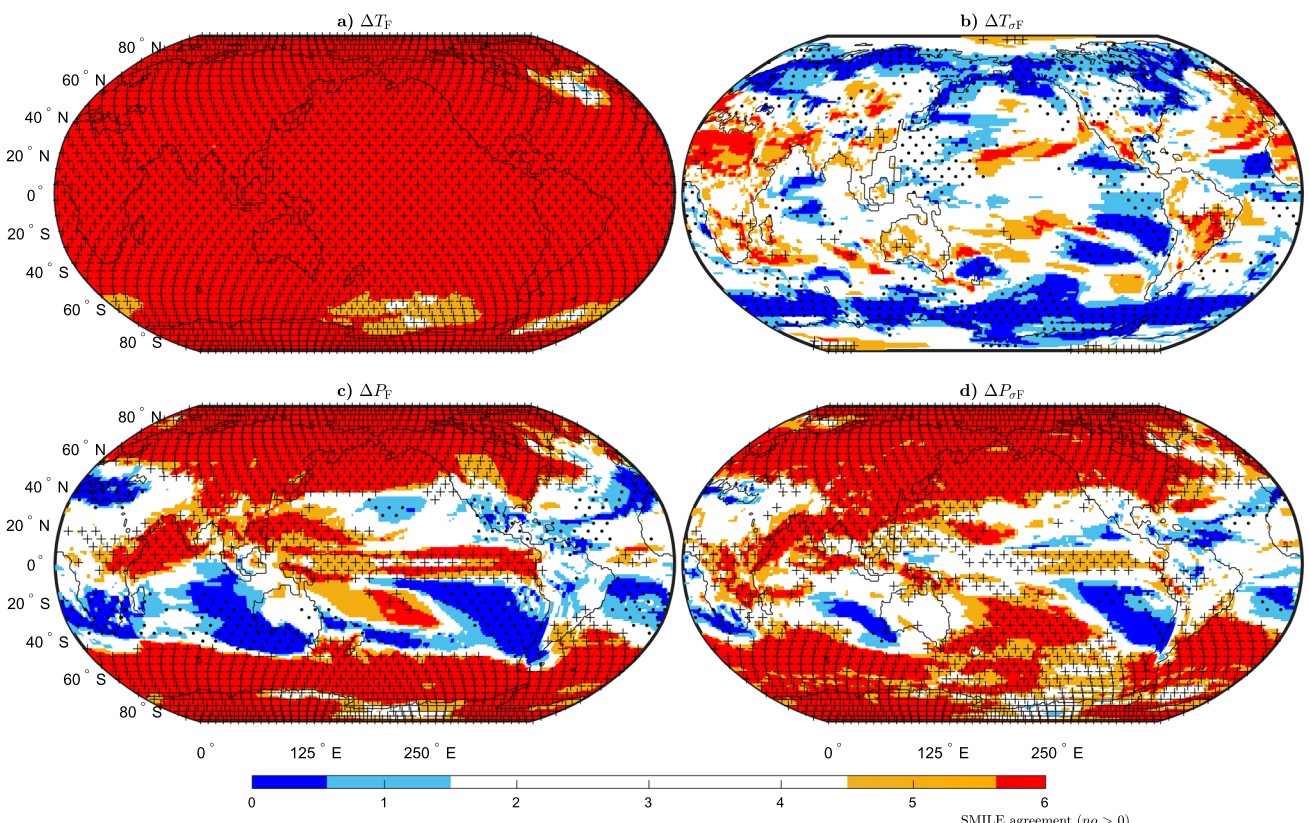

**Fig. 5 Model-to-model agreement on the sign of the response to external forcing. a** Mean-state temperature ($\Delta T_{\mathrm{F}}$), **b** temporal temperature variability ($\Delta T_{\sigma_{\mathrm{F}}}$), **c** mean-state precipitation ($\Delta P_{\mathrm{F}}$) and **d** temporal precipitation variability ($\Delta P_{\sigma_{\mathrm{F}}}$). The forced response is computed for the period 2050–2099 (RCP8.5 forcing) as compared to 1950–1999 (historical forcing). Red shows agreement on an increase in each quantity, while blue shows agreement on a decrease. White regions show <83% agreement on the sign of the change (less than five of six single model initial-condition large ensembles (SMILEs) agree). Stippling shows where there is 79% agreement on the sign of the change using the atmospheric CMIP5 sub-ensembles (11 or more out of the 14 subsets agree), with crosses indicating an increase and dots indicating a decrease. The measures of agreement correspond to a significance level of 0.01 using a binomial distribution.

there is a decrease in $\Delta T_{\sigma_{\mathrm{F}}}$ in both the SMILEs and CMIP5 and Central–South America where both the SMILEs and CMIP5 show a increase in $\Delta T_{\sigma_{\mathrm{F}}}$. There are extended areas of model-to-model agreement in both the SMILEs and CMIP5 showing a decrease in $\Delta T_{\sigma_{\mathrm{F}}}$ in the high latitudes particularly over the Southern Ocean and the Northern Hemisphere high-latitude continental land masses. Where there is model-to-model agreement in the SMILEs, there is often also CMIP5 agreement, particularly in regions where $\Delta T_{\sigma_{\mathrm{F}}}$ decreases. This is, however, not always the case. For example, over Northern Africa, the SMILEs show a large region of agreement on an increase in $\Delta T_{\sigma_{\mathrm{F}}}$ that is not found in CMIP5.

The spatial map of model-to-model agreement on the sign of the forced response is very similar for both $\Delta P_{\mathrm{F}}$ and $\Delta P_{\sigma_{\mathrm{F}}}$. Most locations poleward of 40º show agreement of an increase in both quantities. While there is some agreement between 40ºS and 40 ºN, areas such as most of South America and Australia show no agreement on the sign of the change. There is model-to-model agreement found in large areas of the ocean basins between these latitudes. There are also some differences between the two quantities. While there is an agreement of an increase in $\Delta P_{\mathrm{F}}$ in the tropical Western Pacific, this does not occur for $\Delta P_{\sigma_{\mathrm{F}}}$. We also find much more agreement in the sign of the change over Africa in $\Delta P_{\sigma_{\mathrm{F}}}$ than in $\Delta P_{\mathrm{F}}$. For both $\Delta P_{\mathrm{F}}$ and $\Delta P_{\sigma_{\mathrm{F}}}$ there is almost always CMIP5 agreement of the same sign of change in the same locations as the SMILE agreement. The SMILEs are thus a good

proxy for the CMIP5 archive and we use them in the following sections to delve into the forced changes in three areas that are policy-relevant (i.e., sections of the land surface, the Arctic and the tropical Pacific) to illustrate how the SMILE results can be used.

**An assessment of model-to-model agreement over the land and the Arctic**. We first compute the forced response over the new IPCC-defined regions[46] for Europe, the Arctic, Australia and South-East Asia (Figs. 6 and 7) to determine whether there is model-to-model agreement. We consider only the land surface for all regions, except the Arctic where we consider land, ocean and ice. In regions where five of the six SMILEs agree on the sign of the change, there is a high agreement (83%) in the sign of the change. In regions where all six SMILEs agree, we have a very high agreement.

We first investigate $\Delta T_{\mathrm{F}}$ in Fig. 6a, b. For $\Delta T_{\mathrm{F}}$, we find very high agreement in the sign of the change over all regions considered. However, there is less agreement in the magnitude of the change, which varies between 3 and 4 °C over the three European sectors, up to 10 °C over the Arctic, and 2–3 °C over Australia and South-East Asia. This demonstrates that for $\Delta T_{\mathrm{F}}$, we have an agreement in the sign, but not the magnitude of the change.

When considering $\Delta P_{\mathrm{F}}$ (Fig. 6c, d), we find that all models show a decrease over the Mediterranean, which ranges from a small decrease to 0.4 mm/day. We also find that all models show

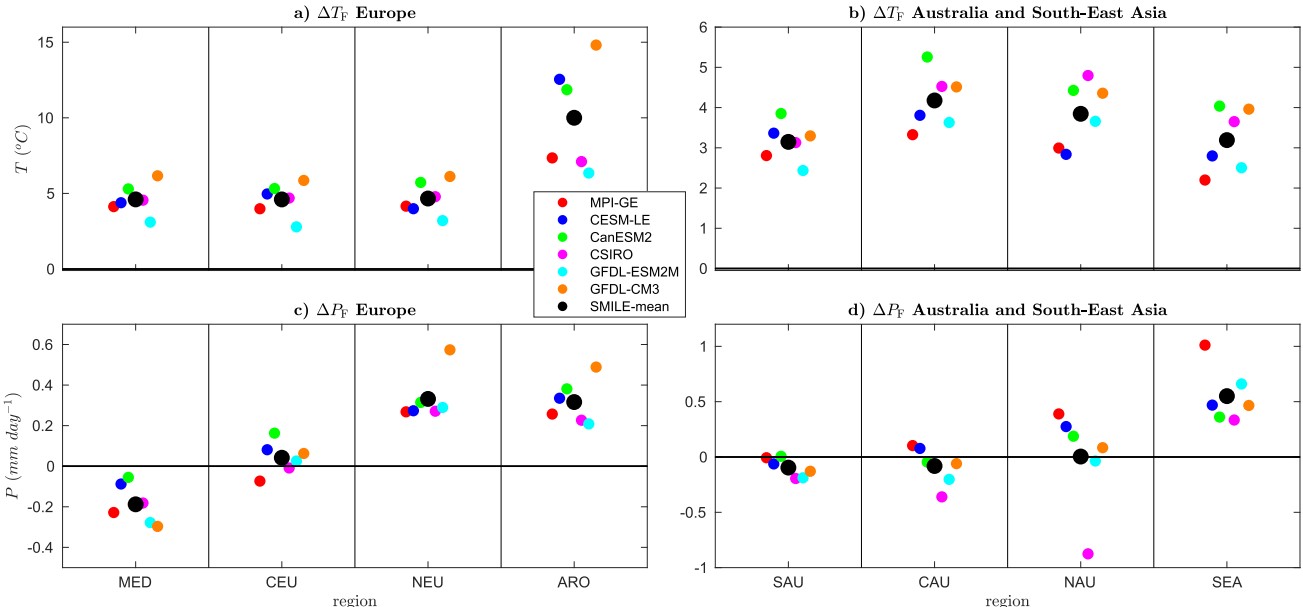

**Fig. 6 Mean-state ensemble-mean single model initial-condition large-ensemble (SMILE) projections of the response of temperature (*T*) and precipitation (*P*) to external forcing for Europe, the Arctic, South-East Asia and Australia.** Forced response over each region in each individual SMILE ($\Delta_{s,F}$, coloured circles) and the SMILE mean ($\Delta_F$, black circle) are shown for the period 2050–2099 (RCP8.5 forcing) as compared to 1950–1999 (historical forcing). The panels show $\Delta_{s,F}$ and $\Delta_F$ for **a** *T* over Europe and the Arctic, **b** *T* over South-East Asia and Australia, **c** *P* over Europe and **d** *P* over South-East Asia and Australia. Error bars are computed by bootstrapping 1000 times with the matlab *bootci* function for the mean. We note that the error bars are very small when compared to the model-to-model differences and are not visible in the figure. Regions are the Mediterranean (MED), Central Europe (CEU), Northern Europe (NEU), Arctic (ARO), Southern Australia (SAU), Central Australia (CAU), Northern Australia (NAU) and South-East Asia (SEA). All regions are defined as in Iturbide et al.[46]. Only the land surface is considered over all regions except ARO, where land, ocean and ice are used.

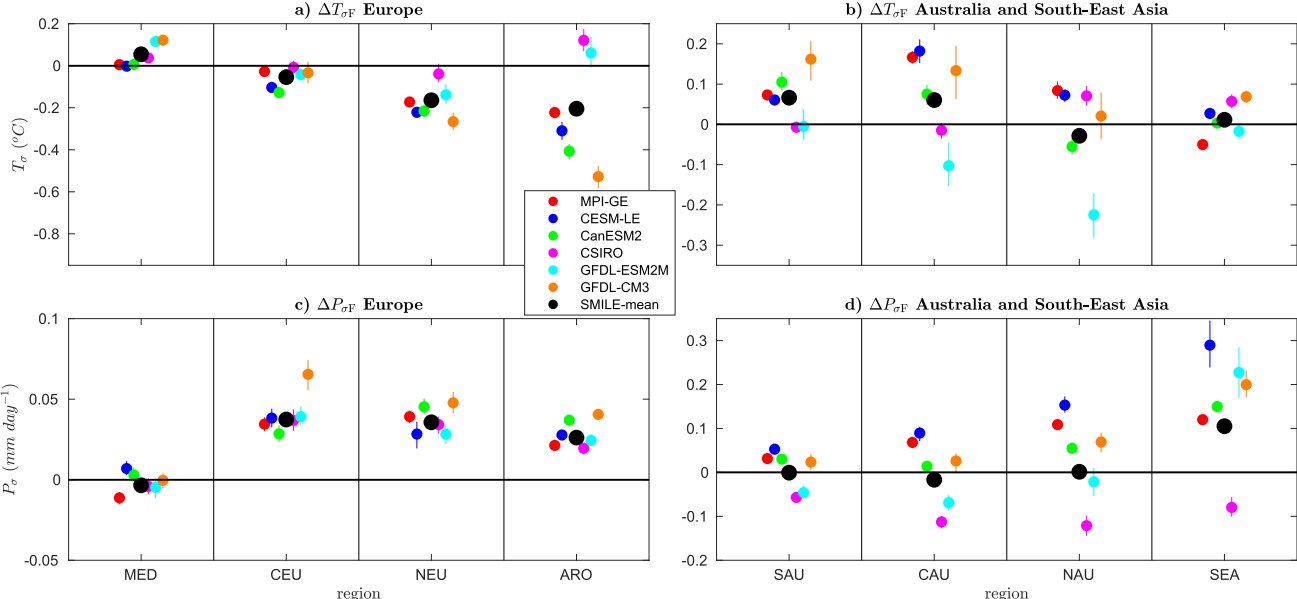

**Fig. 7 Mean-state ensemble-mean single model initial-condition large-ensemble (SMILE) projections of the response of temporal temperature variability (*T$_\sigma$*) and temporal precipitation variability (*P$_\sigma$*) to external forcing for Europe, the Arctic, South-East Asia and Australia.** Forced response over each region in each individual SMILE ($\Delta_{s,F}$, coloured circles) and the SMILE mean ($\Delta_F$, black circle) are shown for the period 2050–2099 (RCP8.5 forcing) as compared to 1950–1999 (historical forcing). The panels show $\Delta_{s,F}$ and $\Delta_F$ for **a** *T$_\sigma$* over Europe and the Arctic, **b** *T$_\sigma$* over South-East Asia and Australia, **c** *P$_\sigma$* over Europe and **d** *P$_\sigma$* over South-East Asia and Australia. Error bars are computed by bootstrapping 1000 times with the matlab *bootci* function for the mean of Eq. (10) (see "Methods"). Regions are the Mediterranean (MED), Central Europe (CEU), Northern Europe (NEU), Arctic (ARO), Southern Australia (SAU), Central Australia (CAU), Northern Australia (NAU) and South-East Asia (SEA). All regions are defined as in Iturbide et al.[46]. Only the land surface is considered over all regions except ARO, where land, ocean and ice are used.

an increase in $\Delta P_F$ over Northern Europe, with all models showing a magnitude of 0.3 mm/day, except GFDL-CM3, which increases by almost 0.6 mm/day. $\Delta P_F$ in the Arctic sector also increases in all models with a range of possible magnitudes (0.2–0.5 mm/day). South-East Asia also has an increase in $\Delta P_F$ in all models, with relative agreement on the magnitude of the change in all models, except MPI-GE. Over Central Europe, all models except MPI-GE show an increase or no change. This is also true for Northern Australia, where all models except CSIRO show an increase or no change. We find that the models agree on either a small decrease or no change in $\Delta P_F$ over Southern Australia, with no agreement over Central Australia.

We find model-to-model agreement on a small increase or no change in $\Delta T_{\sigma_F}$ over the Mediterranean and a small decrease or no change in $\Delta T_{\sigma_F}$ over Central Europe (Fig. 7a, b). All models show a decrease in $\Delta T_{\sigma_F}$ over Northern Europe, with all models decreasing by about 0.2 °C, except CSIRO, which decreases less. There is either no change or an increase in all models over Southern Australia. We find no model-to-model agreement in the sign of $\Delta T_{\sigma_F}$ over the Arctic, Central and Northern Australia and South-East Asia (Fig. 7a, b).

Finally, we consider $\Delta P_{\sigma_F}$ (Fig. 7c, d). All changes in $\Delta P_{\sigma_F}$ over Europe are relatively small (<0.08 mm/day). We find agreement of the increases over Central and Northern Europe, and the Arctic. Over Northern Europe, the magnitude is around 0.04 mm/day in all models, and all models show the same increase over Central Europe except GFDL-CM3. There is no agreement over the Mediterranean and Southern Australia, but the changes are small, so this could indicate agreement of limited change. There is also no agreement over Central and Northern Australia; however, in this case, the differences in magnitude are larger. Finally, over South-East Asia, all models agree on an increase in $\Delta P_{\sigma_F}$ except CSIRO, which shows a decrease.

In general, when all models agree on the sign of the change bar one, the outlying model is CSIRO, with MPI-GE being the outlier for $\Delta P_F$ over Central Europe. However, when all models agree on the magnitude of the change, except one, the outlying model is most often GFDL-CM3. We more often find agreement over Europe and the Arctic than the Australian and South-East Asian regions, with Central and Northern Australia showing the least agreement overall.

**An assessment of model-to-model agreement in the tropical Pacific.** In this section, we examine tropical Pacific projections. In this region, climate models largely agree on projected El Niño-like warming associated with a slowdown of the Walker circulation in the future[47–49]. However, a recent study argues that a La Niña-like warming is physically consistent and occurs in at least one climate model[50]. Furthermore, the earlier assessments[47–49] were based on analyses of multi-model ensembles of opportunity (CMIP3, CMIP5), with only one ensemble member from each model. This begs the question of whether the degree of agreement was over-stated due to a lack of independence in the models considered by chance due to the phase of the internal variability sampled. Using the six SMILEs, we can now delve into where the models agree, and determine which differences are truly due to $U_{MD}$.

Figure 8a, c shows the forced response in the tropical Pacific temperature gradient. While both $\Delta T_F$ and $\Delta P_F$ robustly increase in the tropical Pacific, the gradient across the tropical Pacific does not change consistently across the SMILEs. The SMILE mean shows no change. Four of the SMILEs show an increasing gradient (El Niño-like warming), although this is minimal in two of them, while two show a decreasing gradient (La Niña-like warming). This suggests that there may have been overconfidence in the warming gradient response, which could be due to the use

of too many models that are not independent. On the other hand, it may be that the models used to develop the SMILEs we have analysed do not accurately represent the majority of CMIP3 and CMIP5 climate models. Our results nevertheless indicate that we should not necessarily assume El Niño-like warming and the associated increase in strength of the Walker circulation is correct. Equivalently, our results lower the confidence we have in this aspect of the projections.

Projections of the forced change in the El Niño Southern Oscillation (ENSO) itself in a warming world have become more confident in the recent decade, with many studies showing an increase in $\Delta P_{\sigma_F}$ in the Central-to-Eastern Pacific and a decrease in the far-Western Pacific[28,51]. In addition, a recent study has demonstrated a robust increase in ENSO Eastern Pacific $\Delta T_{\sigma_F}$[30]. However, changes in ENSO can be difficult to assess due to the high internal variability. Indeed, another recent study has shown that a large part of the CMIP5 spread can be replicated using a single SMILE, suggesting that a lot of what has been previously identified as model-to-model differences may just be large internal variability[52].

Figure 8b, d shows the forced response of $\Delta T_{\sigma_F}$ and $\Delta P_{\sigma_F}$ over the full, far West, West, Central and Eastern tropical Pacific. Only the far-Western Pacific and the Eastern Pacific show high model agreement for $\Delta T_{\sigma_F}$, with five models showing a decrease in both regions. This result is at odds with a recent study that shows a more likely increase in ENSO variability in the future[30]. This is likely due to the consideration of different regions, seasons and metrics; however, it suggests that there is still more work needed in this region to reconcile these results. Projections of $\Delta P_{\sigma_F}$ show more agreement, with five of the six models agreeing on an increase or no change in the West, East and Central Pacific. The far-West Pacific shows no model agreement. These results agree well with previous work that finds a more robust increase in the Central and East Pacific than the West[28]. These results can also be seen in Fig. 5, which shows model-to-model agreement of a decrease in $\Delta T_{\sigma_F}$ in the far West and East of the tropical Pacific and an increase in $\Delta P_{\sigma_F}$ in the Central-to-Eastern tropical Pacific.

Previous studies have usually investigated the austral summer season as this is when the largest ENSO variability occurs. When we investigate this season (December, January and February, Supplementary Fig. 12), we find no agreement in any region for $\Delta T_{\sigma_F}$ and now only agreement in the East and Central Pacific for $\Delta P_{\sigma_F}$. When investigating ENSO, it is important to use models that represent ENSO processes well. Three separate studies[30,53,54] have consistently suggested that GFDL-ESM2M and CESM-LE represent ENSO well. However, these two models only agree on the forced response in $\Delta T_{\sigma_F}$ and $\Delta P_{\sigma_F}$ in the far-West Pacific, showing that good performance in simulating the past does not necessarily translate into consistent projections of the future change and highlighting that more work needs to be done in this region to truly understand the future projections from different models.

## Discussion
We have used six SMILEs to quantify the uncertainty due to model-to-model differences and internal variability in projections of temperature, precipitation and their temporal variability under strong forcing in the period 2050–2099 (RCP8.5) as compared to 1950–1999 (historical forcing). The six SMILEs have been pre-viously shown to be largely independent or to behave independently, and they span the CMIP5 model space well. We find that the uncertainty in $\Delta T$ and $\Delta P$ is dominated by $U_{MD}$ similar to previous studies that show that $U_{MD}$ tends to dominate over long timescales[9,10,13]. However, for $\Delta T_\sigma$ and $\Delta P_\sigma$, $U_{MD}$ no longer dominates; $U_{IV}$ is generally larger than $U_{MD}$ in the extratropics,

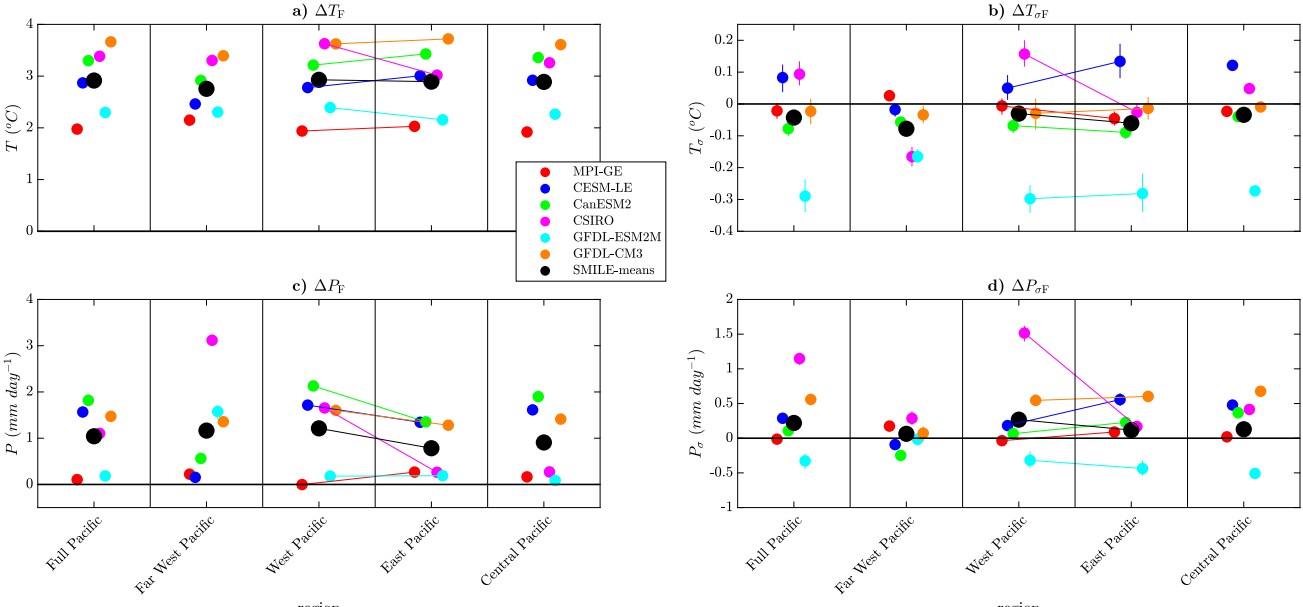

**Fig. 8 Projections of externally forced changes in temperature (*T*), precipitation (*P*) and their temporal variability (*T_σ* and *P_σ*) in the tropical Pacific.** Forced response over the full (160°E–260°E), East (220°E–260°E), Central (190°E–240°E), West (160°E–220°E) and far West (120°E–160°E) Pacific (all between 5°S and 5°N) in each individual single model initial-condition large ensemble (SMILE) ($\Delta_{s,F}$, coloured circles) and the SMILE mean ($\Delta_F$, black circle) are shown for the period 2050–2099 (RCP8.5 forcing) as compared to 1950–1999 (historical forcing). The panels show $\Delta_{s,F}$ and $\Delta_F$ for **a** *T*, **b** *T_σ* **c** *P* and **d** *P_σ*. Horizontal lines are shown between the East and West Pacific to illustrate the proportional change in each variable. Error bars are computed by bootstrapping 1000 times with the matlab *bootci* function for the mean (see a note in "Methods" for $T_\sigma$ and $P_\sigma$ errorbar calculations).

with $U_{MD}$ larger or a similar magnitude to $U_{IV}$ elsewhere. This demonstrates that for temporal variability, an increase in model-to-model agreement may not necessarily decrease the spread of projections on long timescales, in contrast to mean-state temperature and precipitation projections.

We have additionally estimated both $U_{MD}$ and $U_{IV}$ using a multi-model ensemble of opportunity by forming sub-ensembles of CMIP5 models that have a similar atmospheric component. By implementing this new method, which uses the entire CMIP5 archive, we are able to emulate the SMILE results and consistently determine whether $U_{MD}$ or $U_{IV}$ is the dominant source of uncertainty.

By quantifying the size of $U_{MD}$ and assessing the ensemble-mean response to external forcing in each SMILE, we have more accurately identified the extent to which models exhibit robust differences in their response to external forcing as compared to previous studies that used multi-model ensembles. This quantification was not possible in the previous studies, because they were unable to tell whether the models showed different responses due to $U_{MD}$ or $U_{IV}$. For the tropical Pacific, we find that there are model-to-model differences in the sign of the forced response in both $T_\sigma$ in the Central and Western tropical Pacific and in $P_\sigma$ in the far-Western Pacific, although there is more agreement in the other regions of the tropical Pacific.

While we identify regions where there are model differences in the sign of the forced response, we also find extended areas of model-to-model agreement. All models agree that temperature will increase in all regions bar the Northern North Atlantic Ocean and parts of the Southern Ocean. We also find large areas of agreement in both increases and decreases in precipitation and its temporal variability and a high degree of agreement in a decrease of temporal temperature variability in the high latitudes. When considering model-to-model agreement in the sign of the forced change, we find a high degree of agreement between the SMILEs and the CMIP5 atmospheric sub-ensembles, which strengthens these results.

We have assessed the degree of model-to-model agreement not only for the sign of the change, but also for the magnitude. Even

for regions where there is a high degree of agreement on the sign of the change, the magnitude of the externally forced change can vary across models by up to 4 °C over the land surface for $\Delta T_F$ due to $U_{MD}$ alone. These model-to-model differences in magnitude are amplified in the Arctic where the magnitude of the increase in $\Delta T_F$ varies across the models by 10 °C in agreement with previous studies that find large differences in temperature projections in this region[55,56].

The value of the methods used in this paper is in quantifying the uncertainty in both the sign and magnitude of the forced response, as well as determining the spread of what we could observe due to internal variability in the climate system. When using a multi-model ensemble such as CMIP5, we can reasonably estimate the agreement in the sign of the forced change (see the similarity between Fig. 5 and Supplementary Fig. 13); however, we cannot partition the uncertainty into $U_{MD}$ and $U_{IV}$, which means we cannot identify differences in the magnitude of the forced response nor determine how much of the multi-model spread is due to different types of uncertainty. While some estimates of the magnitude of each type of uncertainty can be made using a pre-industrial control for mean-state quantities such as $\Delta T$ and $\Delta P$, this partitioning is not straightforward and involves making assumptions. Indeed, this becomes even more difficult for $\Delta T_\sigma$ and $\Delta P_\sigma$ to the point where it has not even been attempted for these variables. By using SMILEs, we can easily partition the uncertainty and determine what causes the CMIP5 spread for different quantities (Fig. 9).

When considering the example of $\Delta T$ over the globe (Fig. 9, top row), we can see that the CMIP5 distribution can be attributed to $U_{MD}$ alone, consistent with the previous approaches[9]. For $\Delta T_\sigma$ over Northern Europe (Fig. 9, middle row), it becomes clear that the opposite is true and that most of the range of different responses in CMIP5 are due to $U_{IV}$ alone. In this case, the models agree on the magnitude of the forced response; however, using CMIP5 alone, we would not be able to identify this. Finally, the spread in the CMIP5 estimate of South-East Asian $\Delta P_\sigma$ is due to

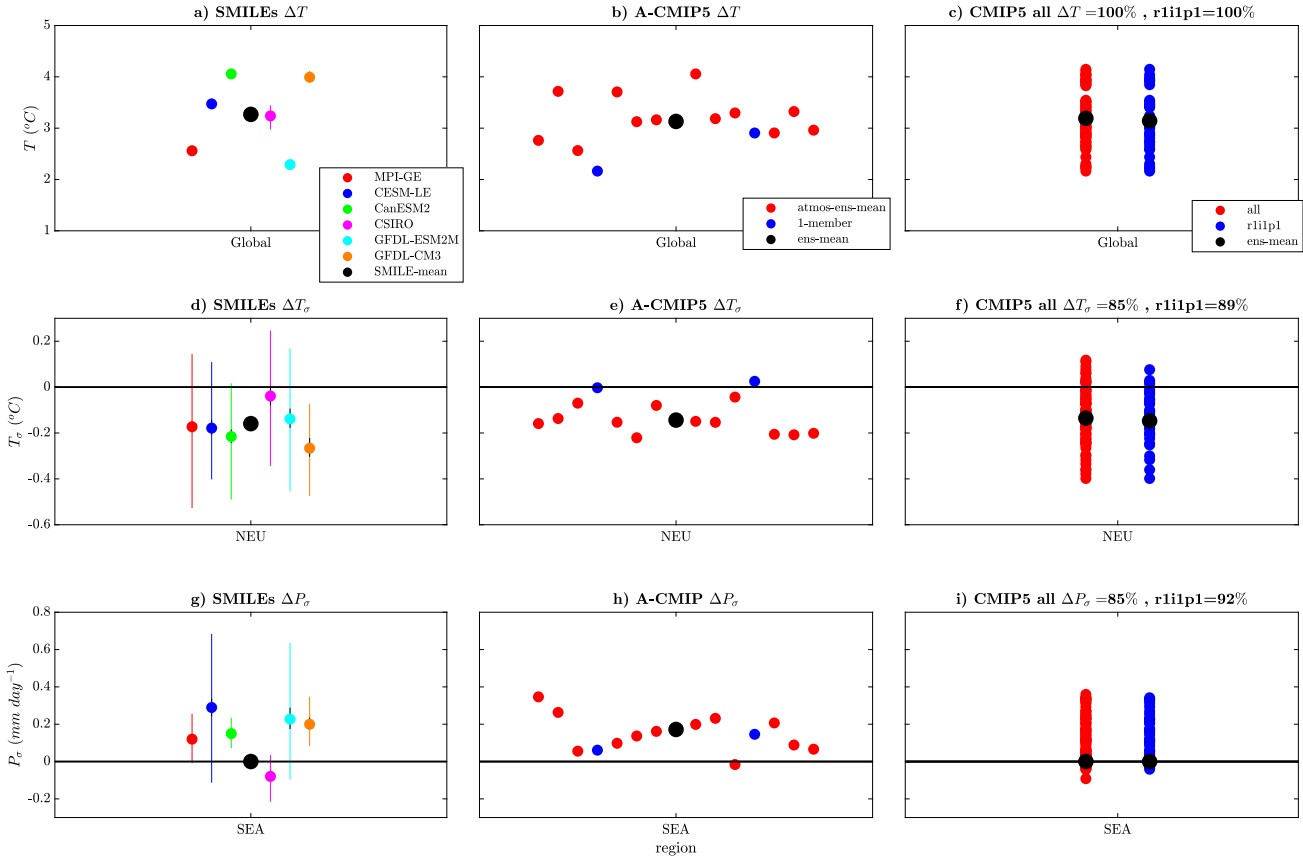

**Fig. 9 The added value of using single model initial-condition large ensembles (SMILEs) and the CMIP5 atmospheric sub-ensembles over a multi-model ensemble. a, d, g** SMILE projections, with individual coloured dots representing each SMILE and the SMILE mean shown in the black dot; uncertainty due to internal variability ($U_{IV}$) is shown in the coloured error bars with the bootstrapped error on the mean shown in the small black error bars (1000 samples using matlab *bootci*). Note that in most cases, the small black error bars are too small to be seen. **b, e, h** Atmospheric subset sub-ensemble projections are shown in the red dots, with the blue dots representing sub-ensembles with only one ensemble member and the black dot the mean taken over all of the sub-ensembles. **c, f, i** CMIP5 multi-model ensemble projections shown for all available ensemble members (red dots), the first member (r1i1p1, blue dots) and the ensemble mean (black dot). Shown for the externally forced response of (**a**–**c**) mean-state temperature ($\Delta T$) global mean, **d**–**f** temporal temperature variability ($\Delta T_\sigma$) over Northern Europe (NEU) and **g**–**i** temporal precipitation variability ($\Delta P_\sigma$) over South-East Asia (SEA). The percentage of models that agree on the sign of the change for CMIP5 is shown in the title of the CMIP5 panels. Only the land surface is used for NEU and SEA. Projections are shown for the period 2050–2099 (RCP8.5 forcing) as compared to 1950–1999 (historical forcing). See a note in "Methods" for $T_\sigma$ and $P_\sigma$ SMILE errorbar calculations.

both $U_{MD}$ and $U_{IV}$. Furthermore, we illustrate a broad consistency of the SMILE results with the atmospheric sub-sampled CMIP5, from which we would draw the same conclusions as the SMILEs. Using both the SMILE and atmospheric sub-ensembles in Fig. 9, we can now identify model-to-model agreement in the magnitude as well as the sign of the forced response, and determine the range of potential observed futures due to internal variability adding important information to the CMIP5 archive traditionally used (Fig. 9, far-right panels).

The partitioning of model uncertainties into $U_{MD}$ and $U_{IV}$ has important implications. $U_{MD}$ is in principle reducible and quantifying it has implications for model development as well as understanding confidence in our projections. By contrast, quantifying the irreducible $U_{IV}$ is important for adaptation purposes, because people need to know the range of outcomes they have to prepare for. The methods presented in this study will remain important as we move to the next generation of climate models.

## Methods

**Models used.** We use six SMILES in this paper. The SMILEs are CanESM2[32], CESM-LE[31], CSIRO-Mk3-6-0[36], GFDL-ESM2M[38], GFDL-CM3[37] and MPI-GE[16] (details in Supplementary Table 1 and Supplementary Figs. 5–8, Deser at al.[33]). We

additionally use all ensemble members from CMIP5 that were available for both historical and RCP8.5 scenarios. For our analysis, we use precipitation (in CMIP5, pr) and surface temperature (in CMIP5, ts) fields (see Supplementary Table 2). We then consider the change between the period 2050–2099 as compared to 1950–1999 using annual-mean data from each model. When considering temporal variability, the data were detrended before calculations. The SMILEs were detrended by removing the ensemble mean at each grid point for each individual month from each ensemble member before the annual means were computed. CMIP5 ensemble members were linearly detrended at each grid point for each 50-year period separately after annual means were computed. Annual means and temporal variability were calculated, then all data was remapped using conservative mapping to a 1° grid before additional analysis and intercomparison.

**Calculation of $U_{IV}$ and $U_{MD}$.** Equations (6)–(8) are similar to those used by Rowell et al.[35]. The correction term (Eq. (8)) occurs because the variance of the ensemble means is a biased estimate of $U_{MD}$ as it still contains an element of internal variability. The larger the ensemble size, the smaller this bias becomes and the smaller the correction term becomes. We include this correction due to the limitation of including some smaller ensembles. We find that the correction term is negligible for T and P and regionally important for ($T_\sigma$) and ($P_\sigma$) (Supplementary Fig. 14).

**Calculation of $T_\sigma$ and $P_\sigma$.** In Eq. (10), we take the difference between standard deviations. This is done because we are interested in the difference in temporal variability between the two time intervals. This differs from the classical statistical

approach as the difference is not based on random variables, but the difference of physical climate parameters in the temporal dimension. When calculating the forced change in temporal variability in Eq. (11), the approach is similar to Eq. (10). However, we need to first calculate the mean internal variability in each time interval. Therefore, we use the square root of the ensemble-mean variance to compute the mean temporal variability in each interval for the entire ensemble. To calculate the multi-ensemble-mean change in Eq. (12), the approach is similar. First, we compute the ensemble-mean variance for each SMILE in each time period. We then average the ensemble-mean variance across the six SMILEs. We then find the multi-ensemble-mean temporal variability in each time period by taking the square root of the multi-ensemble-mean variance.

**CMIP5 sub-ensemble calculations.** The CMIP5 models were subset into groups of models that shared an atmosphere component[6]. These subsets are shown in Supplementary Table 3. The calculation of $U_{MD}$ and $U_{IV}$ was completed by treating these groups as small sub-ensembles and completing the same calculations as for the SMILEs described in the "Results" section. We note that the second correction term for $U_{MD}$ is not used in these calculations. Here $U_{MD}$ is calculated as

$$U_{MD} = \sqrt{D^2} \tag{13}$$

where $D^2$ is the sample variance of the ensemble means as shown in Eq. (7). We tested this sub-ensemble approach using both the atmosphere and ocean components and found similar results (Supplementary Fig. 9). We also tested whether this result could just occur from any type of data subset, by creating random sub-ensembles. We find that the random approach does not work (Supplementary Fig. 9).

**Land, Arctic and Pacific calculations.** Pacific boundaries are as follows: full (160° E–260°E), East (220°E–260°E), Central (190°E–240°E), West (160°E–220°E) and far-West (120°E–160°E) Pacific (all between 5°S and 5°N). The land and Arctic boxes are defined as in Iturbide et al.[46]. For the boxes over land, the ocean is masked using regridded masks on a 1° grid to only include the land surface. Each model is masked using its own separate mask. The Arctic Ocean region (ARO) is not masked, neither are the global changes shown in Fig. 9. When aggregating variability over the different regions, we average the variance and take the square root, similar to ref. [26]. We note that bootstrapping is completed on the mean of Eq. (10), not Eq. (11). We tested whether this approach changes the results, by computing the mean of Eq. (11) and find limited differences. This is exemplified by the fact that the means are surrounded by the bootstrapped estimates.

## Data availability

The data that support the findings of this study are openly available at the following locations: MPI-GE, https://esgf-data.dkrz.de/projects/mpi-ge/, all other large ensembles (CanESM2, CESM-LE, CSIRO-Mk3-6-0, GFDL-ESM2M and GFDL-CM3); http://www.cesm.ucar.edu/projects/community-projects/MMLEA/ and CMIP5, https://esgf-node.llnl.gov/search/cmip5/. Derived data supporting the findings of this study are available at http://hdl.handle.net/21.11116/0000-0007-4AFD-A.

## Code availability

The code used to both process the data and create the figures for this paper can be accessed at http://hdl.handle.net/21.11116/0000-0007-4AFD-A.

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

## Acknowledgements

We thank the US CLIVAR Working Group on Large Ensembles for providing much of the large-ensemble data. We acknowledge the World Climate Research Programme's Working Group on Coupled Modelling, which is responsible for the Coupled Model Intercomparison Project (CMIP), and we thank the climate modelling groups for producing and making available their model output. For CMIP, the US Department of Energy's Program for Climate Model Diagnosis and Intercomparison provides coordinating support and led the development of software infrastructure in partnership with the Global Organization for Earth System Science Portals. We are indebted to Dr. Urs Beyerle for his effort to download and organise the immense CMIP5 dataset. We thank Thibault Tabarin for his helpful comments on the paper as well as Dian Putrasahan, Sebastian Milinski, Dirk Olonscheck, Oliver Gutjahr, Friederike Fröb, Christopher Hedemann, Ruth Lorenz, Flavio Lehner and Angie Pendergrass for their discussion on various parts of the paper. N.M. and J.M. were supported by the Max Planck Society for the Advancement of Science. N.M. was additionally supported by the Alexander von Humboldt Foundation. S.P. is supported by the Australian National Environmental Science Program's Earth System and Climate Change Hub.

## Author contributions

The concept was devised and iterated on in discussions between N.M., S.P. and J.M. N.M. designed and carried out the analysis. N.M. wrote the paper with critical input from S.P and J.M.

## Funding

## Competing interests

The authors declare no competing interests.
