## [Peer Review File · Nature Communications]

Reviewer Comments, first round –

Reviewer #1 (Remarks to the Author):

Maher et al. analyze mean and variability changes in temperature and precipitation using a newly available ensemble of single model initial-condition large ensembles (SMILEs). They highlight some regions where there is agreement in changes in mean or variability, but in many cases (beyond temperature change) there is large disagreement. While interesting, the results are difficult to interpret by themselves without discussion of (1) model fidelity and (2) physical mechanisms. Furthermore, the paper would benefit from additional discussion of statistical significance in both comparing the proposed subsampling of CMIP5 with the SMILEs, and in assessing changes in climate. At this point, I cannot recommend publication.

Major points

- While it is true that the new availability of SMILEs allows for analyses that were not previously possible, they do not erase the need to assess model fidelity. The authors inherently assume that all models in the archive are equally valid representations of reality in both their mean and variance. However, this is unlikely to be true. Without further information about the goodness of each model with respect to the metrics the authors focus on, it is difficult to gain insight from their results.
- One of the key results highlighted in the abstract is a new subsampling method for the CMIP5 archive. While it is reasonable to group models that share components together, it is not a priori obvious that this will be a good way to separate out model differences versus internal variability. As far as I can tell, the only or primary validation the authors present of the method is a visual comparison of the rows in Figs. 1 and 2. I would like to see additional, quantitative validation of the approach before they claim its success. I also found it strange that the authors chose to group CMIP5 models together based on whether they shared an atmosphere or ocean component without further discussion, but then also argued that, although three of the models in the SMILE archive share components, they were somehow found to be sufficient independent to not be grouped together.
- The authors make the assumption beginning with statements on line 44 that the

ensemble mean across an initial-condition large ensemble is a 'precise' estimate of the forced response. This is not generally true: the number of ensemble members needed to average out the variability can be substantial, especially at regional scales. The dots in Figs. 4 and 5 should therefore have uncertainty estimates associated with them, which could be calculated quite simply through bootstrapping ensemble members.

1

- In general, the manuscript is lacking in sufficient details about the methodology. For instance, the term 'variability' is used frequently, but variability can be calculated in many different ways! I imagine that authors are taking the interannual standard deviation, but details like this should be made explicit.
- All results shown in map form cut off at slightly beyond 40 degrees latitude. What is the motivation for not providing results for the full globe?

Minor points

- Line 17: I had to re-read this line multiple times to understand its meaning, because of the large separation between 'contribution' and 'to'. Suggest rewording.
- Line 31: Internal variability is also more important on smaller spatial scales.
- Line 156: Why use surface temperature (ts) rather than the more typical near-surface air temperature (tas)?
- Line 164: In general, average standard deviation should be calculated as the square root of the average across variances. Note that this assumes that a pooled variance is appropriate, which may not be true, since we don't expect the variances across each model to be the same.

Figures

- Figs. 1 and 2 needs labels and units on the colorbars. I also suggest using discrete rather than continuous colorbars for clearer interpretation.
- Fig. 3 should indicate sign of the agreement via different types of stippling if not always consistent with SMILES.

Reviewer #2 (Remarks to the Author):

Review on 'Robust assessment of model-to-model agreement in projections of future climate' by N. Maher et al.

The manuscript makes use of a new archive of single model initial condition large ensemble (SMILES), to better quantify the role of internal variability in projections of future change. Focus is given to temperature and precipitation in the tropical Pacific region and several ENSO-related

changes are discussed in more detail in the second part of the study.

I find the manuscript to be well written and concise but quite descriptive at several points and believe that it would benefit from including some additional statistics and/or concrete numbers at several places. In addition there are many issues that need to be resolved such as several quite generalizing statements, missing units in several figures, and many instances of lacking preciseness in the description of the figures as pointed out in the minor comments below.

In general I think that the manuscript has the potential of being valuable and interesting contribution to the field, in particular in pointing out the importance of internal variability.

Minor comments

L12-14: I do not think that this statement is valid in such a general form. For a 20-year average of global temperature, for example, I would assume that we have quite a good handle on internal variability.

L14-15: Model inter-dependence does not seem like a 'fundamental limit' in the evaluation of the CMIP MMEs to me, in fact many studies have suggested mechanisms to account for it, among them the ones cited by the authors here.

L22: "SMILEs"

L23: "SMILEs"

L23-24: I assume the authors refer to changes from historical to some end-of-century period based on some emission scenario?

L32: The authors could also mention here that the role of internal variability is also highly dependent on the spatial aggregation and the variable.

L34-36: As mentioned above this is only valid if it is not accounted for.

L38-48: I feel that this paragraph contains a number of vague statements which I would argue is partly due to the lack of a rigorous definition of 'large ensemble'. I agree with the authors that a 100 member SMILE can probably safely be termed 'large' but for a 20 member SMILE I'd like to see some additional arguments.

E.g., I could imagine a criterion being that SMILEs should accurately isolate the forced response (based on some test metric) which the authors state as given: 'the mean across the ensemble provides a precise estimate of each individual SMILE's response to external forcing' (l44-45). Is this really true for a 20 member ensemble already and does it hold across different variables and spatial aggregations (I imagine there being vast differences between, e.g., global annual mean temperature and regional seasonal precipitation)?

CMIP5 already contained several models with up to 10 ensemble members, I'd be hesitant to generalize that the SMILE 'ensemble sizes are much larger than those' (l41-42). In addition there are several studies that have made use of this ensemble members and not just used one member per model as the authors seem to indicate.

L58: While it is described in the Methods section, I think it would be helpful for the reader if the authors mentioned once that they are considering RCP85 in their study.

Figures:

- Several figures miss units
- I'm not sure I understand the cross-like shape on the plots. It seems to indicate 0 lat/lon but why does the horizontal line not extend over the full figure? I would probably find it more helpful to have the pacific regions indicated in one of the plots instead (or in addition)
- The authors could consider adding some lat/lon ticks to figures 1 and 2
- The ticks for lat/lon in figure 3 are missing

- In the method section the authors use the -180/180 notation while figure 3 uses 0/360 please make this consistent.

L62: The role of the different ensemble sizes should be discussed here. How strongly do the SMILEs differ in their spread and does this spread correlate with the ensemble size or is it rather a model property?

L64-67: I have to admit I find it a bit hard to follow here. The authors generally refer to the tropical Pacific as an example region where the internal variability dominates. However, this seems to be only true for the variability changes, correct? Maybe the authors could find a way of showing relative contribution in the plots pair b-c, e-f, etc? Similar to Hawkins 2009? This would more clearly highlight the regions where internal variability dominates.

L80-81: In fact model differences in temperature variability seem to be equally large in the tropical Pacific region, correct?

L85: The authors have described the main features visible in figures 1 & 2 in the last two paragraphs but leave me wondering: What do these results mean? Are they surprising or expected? What have we learned from the usage of these SMILEs (which where computationally very expensive to create I imagine) that we could not have learned otherwise?

Figure 3: I find the labeling of the colorbar counter-intuitive. 0 models agree (darkest blue) seems to indicate that the models do not agree at all when it fact it means that all models agree on a cooling, correct?

Is it possible to also differentiate between agreement on increase/decrease for the CMIP models (e.g., by using different hatching)?

This is a minor point, but is there one model that shows cooling at a grid cell near Bangladesh?

L91-92: 'Many areas of the land surface show agreement on an increase in variability' This seems to be true only for the SMILEs. Can the authors specify how much of the land area shows an increase for either case?

L93-94: 'with large areas of agreement in other regions.' Maybe: 'with large area of agreement on either a decrease or an increase in other regions'

L94-95: Can the authors try to be more precise here? Europe is not really shown in figure 3 so I can not check but there are large parts of the middle east where I do not see any model agreement neither for SMILEs nor for CMIP5. In turn, there are large regions in south Africa where SMILEs as well as CMIP5 agree on a decrease in precipitation.

L98: 'We find that in general the CMIP5 agreement occurs in the same locations as the SMILE agreement.' Can the authors make some more precise statement here? E.g., for temperature variability I would tend to disagree.

L106-108: 'We find very high confidence that temperature variability increases over Africa, and high confidence that it increases over Europe, with an increase or no change over north America and Australia.' Figure 4b looks to me as if there should be high confidence also for north America and possibly also Australia? Granted the change for north America is small but still SMILEs seem to mostly agree.

L109-111: 'Precipitation variability is, however, projected to increase over North America, Africa and Asia with high confidence and over Europe with very high confidence.' Similar to my last comment, I find this to not fully reflect what I see in figure 4d. CSIRO for north America seems to be above the zero line, so the confidence according to the definition by the authors should be very high. But again there is hardly any change in this region, so maybe it should rather be pointed out that there is agreement on no change?

L112: 'This is particularly true for temperature projections over all regions.' Why is this not true for

precipitation change? One could argue that given the mean signal the relative spread is even larger for precipitation.

L121: 'Using the six SMILEs we can now reconcile these conflicting results.' From reading the paragraph I'm not sure what the authors mean by reconcile. The authors suggest that the earlier results may have been overconfident due to model inter-dependence but this remains speculative (or otherwise I can not follow the argument). There is no discussion of underlying reasons for the discrepancy between models, neither with regard to the SMILEs used here nor with regard to the models used in the other two studies. However, this would in my opinion be necessary to truly 'reconcile' this conflict.

L122-123: 'We find that, while both precipitation and temperature are robustly projected to increase in the Pacific' Add 'tropical Pacific'?

I realize that the authors define the "Pacific" as only reaching from -5 to 5 N in the methods section but this is not a generally agreed definition of the Pacific so I would strongly suggest to be clear in the language here in order to not confuse the reader.

L123-124: 'Although the SMILE mean shows an increasing gradient' Are the authors referring to the black line between west and east Pacific in figure 5a here? I'm not able to detect any slope in this line.

L135-136: 'The change in precipitation variability in the far western Pacific varies widely among models' This does not seem to reflect figure 5d, where in fact the model spread in the Far West Pacific is smallest?

L144: "non-independent groups" This seems a bit ambiguous to me. The models within a group are "non-independent" but the groups are then considered independent, correct?

L157: I have to admit I'm a bit surprised by this. Why are the authors using a 1 degree grid? I would assume that this means effectively down-scaling most of the models, correct?

L203: 'Nature'

Reviewer #3 (Remarks to the Author):

Review of Maher et al. 2019

In this paper, the authors present a method to evaluate robustly model-to-model agreement in climate projections based on Single Model Initial-conditions Large Ensembles (SMILEs) and describe its application to some specific cases. The method in itself is not new, but it was not possible to use it before in practice because ensembles of SMILEs did not exist. The authors also describe an alternative new method that can be used when only one or a few members are available for each climate model, based on the fact the climate models are often not independent. They use simulations from "similar" models as members of the same model. This method has some limitations that should better be acknowledged.

The paper is clear, the results are sound and interesting. They are however not especially "cutting-edge" and I wonder whether it is justified to publish this work in a high-impact journal, but it is more a question for the editorial team. It is true that it is the first or one of the first papers to use a significant number of SMILEs to assess model-to-model agreement, but I also have the feeling that they could apply this approach to more interesting cases. And the number of SMILEs used (6) is still very small: it is necessary to remain cautious with the results shown. That being said, I think the paper needs moderate revisions before being publishable.

Major comments.

-The focus of the paper is not always clear. At first, we have the impression that the paper is mainly methodological but then, an important part is dedicated to some applications. But the authors don't go very "deep" in these applications.

It is not really a problem, but for example I wonder whether the introduction is totally adapted to the actual content of the paper.

-A better discussion about the existing ways to characterize model agreement is needed, I think, as several methods have been proposed (e.g. the discussion in the IPCC AR5). For example, it is possible to characterize "robustly" model agreement with a single member per model and piControl simulations: we can characterize whether the models agree on a significant (greater than internal variability) change. The approach of the authors allows to show model agreement even if the future change is smaller than internal variability for individual models. This should be discussed, I think.

Also, the question of model agreement basically lies on the one of the estimation of the forced signal and different methods can be used in that context. It is possible to do the same analyses as in this paper but with a different method to estimate the forced signal (e.g. spline, loess-filter etc. for example). Some authors have used this approach (e.g. Hawkins and Sutton 2009) to separate internal variability and the forced signal with a single member per model. I personally think that this approach has major limitations, and that using SMILEs to estimate the signal is the only method that is really satisfactory.

But I think that this point should be discussed, in order for the authors to convince all readers of the interest of their approach.

-The study area looks strange to me. Why do the authors stop their maps at 40° N and S? It would be possible to show global maps or at least maps with mid-latitudes. It is all the stranger because the authors discuss the results for Europe, but only the extreme south of Europe is visible on the maps. Spatial averages over "Europe" are shown but Europe is defined as 50°E – 150 ° E and 20° N – 40°N. None of these latitudes / longitudes makes sense to define Europe. With this definition, India is at the center of Europe ! Maybe the authors meant -50°E but even in that case the latitudes do not make sense. Cairo is not in Europe. Such weird definition of regions is also seen for North-America, whose definition excludes a large part of USA, and Canada in totality. This is really misleading.

I have the feeling (based on what is visible for the northern part of the domain in Figure 1e) that the authors may use their peculiar study area because their approach based on the re-sampling of CMIP5 models does not work at higher latitudes. Is that the case? Why not simply say it (and show it)?

-I think the authors overestimate the interest of their approach to estimate model-to-model agreement based on the re-sampling of single (or few) members from CMIP5 models that share a common atmospheric / oceanic model. It is true that two models that share some common components should not be considered independent, but it is also too extreme to consider that two coupled climate models that share the same atmospheric model are in fact the same model and that their differences are only due to internal variability. And it is clearly visible in the results of some analyses of the paper. The role of "internal variability" is often largely greater in Figure 1e than in 1b (even if it is impressive to see the differences between Figure 1e and 1f). This should be acknowledged.

I don't suggest to remove these analyses, which are interesting, but I think the discussion needs to be more balanced, and the limitations need to be discussed: the method can give an indication but it is far from perfect etc...

I also think it would be interesting to better illustrate how the lack of independence impacts model agreement, e.g. compare CMIP5 model agreement with their approach and the classical approach (based on the hypothesis that all models are independent).

Minor comments

L16-17 "largely independent": please describe the models somewhere, with the components etc. How GFDL-CM3 and GFDL-ESM2M can be said to be independent? They have a lot in common, haven't they?

L19-21. Not clear. I had to read the whole manuscript to understand what is meant here.

L26. What has been done wrong in 7 and 8 ?

L51-52. OK. But the reference 15 only deals with precipitation biases over the South Asian. It is not sufficient to substantiate the affirmation (however, it is true that looking at Figure 5, GFDL-ESM2M and GFDL-CM3 do seem to behave differently).

L62. Please refer to the method section here and where necessary. At first, I wondered how some analyses were done, whereas they are explained in the Method section.

L65-67. In the Tropical Pacific, the internal variability is not especially large in Figure 1b,e. Do you talk about precipitation or temperature variability change? Please be more precise.

L66. "even if the models agree on projected change". Based on which analysis? Or on which references? Do you refer to your analysis in Figure 5? In that case, say it, please. But as shown in Figure 5, models do not agree on precipitation variability change. So you have to be more precise.

L77. "Accurately" is quite exaggerated to my opinion. It is interesting and it is true that it is quite spectacular to see how models from the same pool behave similarly compared to the other models (Figure 1e versus 1f). But we can still see that the "internal variability change" with this approach (Figure 1e) is largely greater than the one estimated with SMILEs, maybe two times greater, even if the color palette does not make this estimation easy.

L89 and Figure 3. You could use two symbols in Figure 3, to show where the CMIP5 models agree on an increase and where they agree on a decrease. Is there at some places an agreement in the SMILEs on an increase and an agreement in the CMIP5 models on a decrease?

L94. See one of my major comment.

L119-120. It would be interesting to further discuss the differences between the results of 23 and 24. Is it the way the models are analyzed? Is it because the two studies use different samples of models? We can guess from lines 121 to 127 but a sentence would be nice.

L141-142. "For the first time...". Arguably Hawkins and Sutton (2009) already did that. I would say that it is done for the first time "correctly", without to have to use an approximate estimate of the forced signals.

L150. Agreed. You could add a word of caution about the limitations to the results in this paper due to the small number of SMILEs available today.

L160-161. Given the approach proposed by the authors to assess agreement with CMIP5 models would it not been more logical and consistent to detrend the CMIP5 models using the ensemble means within each group?

Reviewers' comments:

Reviewer #1 (Remarks to the Author):

Review of 'Robust assessment of model-to-model agreement in projections of future climate'

Summary

Maher et al. analyze mean and variability changes in temperature and precipitation using a newly available ensemble of single model initial-condition large ensembles (SMILEs). They highlight some regions where there is agreement in changes in mean or variability, but in many cases (beyond temperature change) there is large disagreement. While interesting, the results are difficult to interpret by themselves without discussion of (1) model fidelity and (2) physical mechanisms. Furthermore, the paper would benefit from additional discussion of statistical significance in both comparing the proposed subsampling of CMIP5 with the SMILEs, and in assessing changes in climate. At this point, I cannot recommend publication.

We thank the reviewer for their helpful comments and have substantially revised the manuscript to address them. We address the individual points from this initial summary where they are raised individually below.

Major points

- While it is true that the new availability of SMILEs allows for analyses that were not previously possible, they do not erase the need to assess model fidelity. The authors inherently assume that all models in the archive are equally valid representations of reality in both their mean and variance. However, this is unlikely to be true. Without further information about the goodness of each model with respect to the metrics the authors focus on, it is difficult to gain insight from their results.

The results in this paper address four quantities across the globe. Quantifying and assessing model fidelity at each grid point this is out of the scope of this paper. Model-to-model agreement without considering model fidelity is the usual method of choice to assess uncertainty, as witnessed by AR5 Ch12 and many others (see discussion on lines 60-68). Additionally there is no single approach to assessing model weighting for fidelity or independence (see lines 52-58). In this study by using SMILEs we can assess model-to-model agreement, without the influence of internal variability, by isolating the forced change in each SMILE as shown in equation 1 and build on previous assessments of model-to-model agreement by doing so.

While assessing model fidelity at each grid point is not within the scope of the paper we identify which models are generally the outliers on lines 290-293 and assess how the results are affected by model fidelity in the tropical Pacific as there has been much previous work in this region identifying which models can represent ENSO well (see lines 323-327).

- One of the key results highlighted in the abstract is a new subsampling method for the CMIP5 archive. While it is reasonable to group models that share components together, it is not a priori obvious that this will be a good way to separate out model differences versus internal variability. As far as I can tell, the only or primary validation the authors present of the method is a visual comparison of the rows in Figs. 1 and 2. I would like to see additional, quantitative validation of the approach before they claim its success. I also found it strange that the authors chose to group CMIP5 models together based on whether they

shared an atmosphere or ocean component without further discussion, but then also argued that, although three of the models in the SMILE archive share components, they were somehow found to be sufficient independent to not be grouped together.

We now validate and discuss the differences between the CMIP5 method and the SMILE method on lines 200-226 and have added Supplementary Figure S10, which shows the differences between the two methods for the entire globe. Should the reviewer find it appropriate would be happy to include Figure S10 in the main manuscript.

Ideally we would only use SMILEs that are completely independent, however this is not possible. We now address this point in detail on lines 123-136.

- The authors make the assumption beginning with statements on line 44 that the ensemble mean across an initial-condition large ensemble is a ‘precise’ estimate of the forced response. This is not generally true: the number of ensemble members needed to average out the variability can be substantial, especially at regional scales. The dots in Figs. 4 and 5 should therefore have uncertainty estimates associated with them, which could be calculated quite simply through bootstrapping ensemble members.

We have added bootstrapped uncertainty to Figures 4 and 5 (now Figures 6, 7, & 8). In general the uncertainty is low due to the ensemble size and the large area averages. The ensemble size issue is also now discussed on lines 111-121 and in the Supplementary material in detail. We have also added Supplementary Figures S1-S4 to investigate this issue.

- In general, the manuscript is lacking in sufficient details about the methodology. For instance, the term ‘variability’ is used frequently, but variability can be calculated in many different ways! I imagine that authors are taking the interannual standard deviation, but details like this should be made explicit.

More details of the methodology are added up front on lines 94-109, where we now use equations to explain how the different terms used in the paper were calculated. We have also revised the manuscript to be careful about our use of the word ‘variability’.

- All results shown in map form cut off at slightly beyond 40 degrees latitude. What is the motivation for not providing results for the full globe?

We have now extended the analysis to include the entire globe.

Minor points

- Line 17: I had to re-read this line multiple times to understand its meaning, because of the large separation between ‘contribution’ and ‘to’. Suggest rewording.

This sentence has been reworded to read “Here, using a new archive of six largely independent single model initial-condition large ensembles (SMILEs) we can simply separate the contribution of UMD and UIV in the response to strong forcing (RCP8.5) for projections of temperature (ΔT), precipitation (ΔP) and for the first time their temporal variability (ΔT_{var} & ΔP_{var})” (see lines 14-17).

- Line 31: Internal variability is also more important on smaller spatial scales.

We have added this to line 41.

- Line 156: Why use surface temperature (ts) rather than the more typical near-surface air temperature (tas)?

We choose to use ts for this analysis for consistency with Power et al 2013 who also use surface temperature as this was the original motivation for this study.

- Line 164: In general, average standard deviation should be calculated as the square root of the average across variances. Note that this assumes that a pooled variance is appropriate, which may not be true, since we don't expect the variances across each model to be the same.

This is correct, if we assume that U_{MD} and U_{IV} are gaussian and uncorrelated. However, the comparison of the two quantities works irrespective of how they are added to the total, so this should not affect the analysis.

Figures

- Figs. 1 and 2 needs labels and units on the colorbars. I also suggest using discrete rather than continuous colorbars for clearer interpretation.

We choose to keep the colorbars continuous for Figures 1 and 2 (now Figures 1-4) as this emphasises the patterns that we see. We have, however, added a fourth panel to help with interpretation, which has discrete colorbars. We have added units on the colorbars as requested.

- Fig. 3 should indicate sign of the agreement via different types of stippling if not always consistent with SMILES.

We have added the sign of the CMIP5 change by using two types of stippling.

Reviewer #2 (Remarks to the Author):

Review on 'Robust assessment of model-to-model agreement in projections of future climate' by N. Maher et al.

The manuscript makes use of a new archive of single model initial condition large ensemble (SMILES), to better quantify the role of internal variability in projections of future change. Focus is given to temperature and precipitation in the tropical Pacific region and several ENSO-related changes are discussed in more detail in the second part of the study.

I find the manuscript to be well written and concise but quite descriptive at several points and believe that it would benefit from including some additional statistics and/or concrete numbers at several places. In addition there are many issues that need to be resolved such as several quite generalizing statements, missing units in several figures, and many instances of lacking preciseness in the description of the figures as pointed out in the minor comments below.

In general I think that the manuscript has the potential of being valuable and interesting contribution to the field, in particular in pointing out the importance of internal variability.

We thank the reviewer for their helpful comments.

We have substantially revised the manuscript to be much more precise and have added units to all Figures where they were missing except for Figure 5, where we describe the colorbar in the caption.

Minor comments

L12-14: I do not think that this statement is valid in such a general form. For a 20-year average of global temperature, for example, I would assume that we have quite a good handle on internal variability.

This sentence no longer exists, however, we now use the words ‘difficult’ and ‘limited’ in the manuscript instead of ‘cannot’ to address this concern.

L14-15: Model inter-dependence does not seem like a ‘fundamental limit’ in the evaluation of the CMIP MMEs to me, in fact many studies have suggested mechanisms to account for it, among them the ones cited by the authors here.

This is true, however it mechanisms to account for this model inter-dependence are usually only used in ensemble mean evaluations, not when partitioning uncertainties. We now discuss previous methodologies in more detail on lines 52-78.

L22: “SMILES”

This has been fixed

L23: “SMILES”

This has been fixed

L23-24: I assume the authors refer to changes from historical to some end-of-century period based on some emission scenario?

This information has been added on lines 16 and 100.

L32: The authors could also mention here that the role of internal variability is also highly dependent on the spatial aggregation and the variable.

This line has been revised to include this statement (lines 40-44).

L34-36: As mentioned above this is only valid if it is not accounted for.

This is often not accounted for. We have extended the introduction to better summarise the previous work. See lines 52-78.

L38-48: I feel that this paragraph contains a number of vague statements which I would argue is partly due to the lack of a rigorous definition of ‘large ensemble’. I agree with the authors that a 100 member SMILE can probably safely be termed ‘large’ but for a 20 member SMILE I’d like to see some additional arguments.

E.g., I could imagine a criterion being that SMILES should accurately isolate the forced response (based on some test metric) which the authors state as given: ‘the mean across the ensemble

provides a precise estimate of each individual SMILE's response to external forcing' (144-45). Is this really true for a 20 member ensemble already and does it hold across different variables and spatial aggregations (I imagine there being vast differences between, e.g., global annual mean temperature and regional seasonal precipitation)?

CMIP5 already contained several models with up to 10 ensemble members, I'd be hesitant to generalize that the SMILE 'ensemble sizes are much larger than those' (141-42). In addition there are several studies that have made use of this ensemble members and not just used one member per model as the authors seem to indicate.

We now discuss the role of ensemble size on lines 111-121 and in detail in the Supplementary material and have added Figures S1-S4 to address this point. We also discuss in the introduction that although CMIP5 has some ensembles, often only one member is used. See lines 70-78.

We have removed the statement that ensemble sizes are much larger. Instead we refer to the SMILE archive, which is referenced in Deser et al, 2020.

We additionally have added bootstrapped errorbars to Figures 6-8.

L58: While it is described in the Methods section, I think it would be helpful for the reader if the authors mentioned once that they are considering RCP85 in their study.

This information has been added on lines 16 and 100.

Figures:

- Several figures miss units

Units have been added to all Figures bar Figure 5, where we describe the color scheme in the caption.

- I'm not sure I understand the cross-like shape on the plots. It seems to indicate 0 lat/lon but why does the horizontal line not extend over the full figure? I would probably find it more helpful to have the pacific regions indicated in one of the plots instead (or in addition)

The cross has been removed.

- The authors could consider adding some lat/lon ticks to figures 1 and 2

We have not done this, but added labeling for the latitudes and longitudes.

- The ticks for lat/lon in figure 3 are missing

We have not done this, but added labeling for the latitudes and longitudes.

- In the method section the authors use the -180/180 notation while figure 3 uses 0/360 please make this consistent.

We have fixed this to be consistently 0/360.

L62: The role of the different ensemble sizes should be discussed here. How strongly do the SMILEs differ in their spread and does this spread correlate with the ensemble size or is it rather a model property?

This is now discussed on lines 111-121 and in the Supplementary material.

164-67: I have to admit I find it a bit hard to follow here. The authors generally refer to the tropical Pacific as an example region where the internal variability dominates. However, this seems to be only true for the variability changes, correct?

Maybe the authors could find a way of showing relative contribution in the plots pair b-c, e-f, etc? Similar to Hawkins 2009? This would more clearly highlight the regions where internal variability dominates.

We have added a fourth column to Figures 1 and 2 (now Figures 1-4) to show the relative contribution of internal variability and model-to-model differences. We have additionally significantly revised the text to be more precise and clear in this section.

L80-81: In fact model differences in temperature variability seem to be equally large in the tropical Pacific region, correct?

We have revised the text to be preciser and clearer based on the Figures and the new fourth column panels.

L85: The authors have described the main features visible in figures 1 & 2 in the last two paragraphs but leave me wondering: What do these results mean? Are they surprising or expected? What have we learned from the usage of these SMILEs (which were computationally very expensive to create I imagine) that we could not have learned otherwise?

We now address the implications of the results in Figure 1 & 2 (now Figures 1-4) on lines 164-169 and 192-197. We also include Figure 9, which demonstrates how this analysis adds to what would have been found using CMIP5 traditional methods alone (lines 358-373). We have also substantially revised/rewritten the discussion to address this point (see lines 326 onwards).

Figure 3: I find the labeling of the colorbar counter-intuitive. 0 models agree (darkest blue) seems to indicate that the models do not agree at all when in fact it means that all models agree on a cooling, correct?

This is correct. We choose to leave this colorbar as is as red indicates agreement on warming, blue indicates agreement on cooling and white indicates no agreement. This is made clear in the text on line 230-232 and in the caption.

Is it possible to also differentiate between agreement on increase/decrease for the CMIP models (e.g., by using different hatching)?

Yes we have revised the Figure to do this.

This is a minor point, but is there one model that shows cooling at a grid cell near Bangladesh?

Yes there is. Additionally we also now identify the North Atlantic warming hole and regions of the Southern Ocean where temperature is not increasing in all models.

L91-92: 'Many areas of the land surface show agreement on an increase in variability' This seems to be true only for the SMILEs. Can the authors specify how much of the land area shows an increase for either case?

This statement was imprecise and has been removed.

L93-94: 'with large areas of agreement in other regions.' Maybe: 'with large area of agreement on either a decrease or an increase in other regions'

The text has been revised and this sentence no longer exists.

L94-95: Can the authors try to be more precise here? Europe is not really shown in figure 3 so I can not check but there are large parts of the middle east where I do not see any model agreement neither for SMILEs nor for CMIP5. In turn, there are large regions in south Africa where SMILEs as well as CMIP5 agree on a decrease in precipitation.

We have revised the text and extended the Figure to include the entire globe.

L98: 'We find that in general the CMIP5 agreement occurs in the same locations as the SMILE agreement.' Can the authors make some more precise statement here? E.g., for temperature variability I would tend to disagree.

We have revised the whole section to make clearer and more precise statements.

L106-108: 'We find very high confidence that temperature variability increases over Africa, and high confidence that it increases over Europe, with an increase or no change over north America and Australia.' Figure 4b looks to me as if there should be high confidence also for north America and possibly also Australia? Granted the change for north America is small but still SMILEs seem to mostly agree.

We have changed the regions used in Figure 4 (now Figures 6 & 7) and have completely rewritten the section to be much more precise.

L109-111: 'Precipitation variability is, however, projected to increase over North America, Africa and Asia with high confidence and over Europe with very high confidence.' Similar to my last comment, I find this to not fully reflect what I see in figure 4d. CSIRO for north America seems to be above the zero line, so the confidence according to the definition by the authors should be very high. But again there is hardly any change in this region, so maybe it should rather be pointed out that there is agreement on no change?

We have changed the regions used in Figure 4 (now Figures 6 & 7) and have completely rewritten the section to be much more precise. We have additionally added discussion around agreement on limited and no changes in the section beginning on line 255.

L112: 'This is particularly true for temperature projections over all regions.' Why is this not true for precipitation change? One could argue that given the mean signal the relative spread is even larger for precipitation.

This line no longer exists, although we still use temperature as an example of model agreement in the sign of the change, but not the magnitude.

L121: 'Using the six SMILEs we can now reconcile these conflicting results.' From reading the paragraph I'm not sure what the authors mean by reconcile. The authors suggest that the earlier results may have been overconfident due to model inter-dependence but this remains speculative (or otherwise I can not follow the argument). There is no discussion of underlying reasons for the

discrepancy between models, neither with regard to the SMILEs used here nor with regard to the models used in the other two studies. However, this would in my opinion be necessary to truly ‘reconcile’ this conflict.

We have revised this paragraph on lines 296-309. We no longer use reconcile, but explain the conflict and what we do to address it in more detail.

L122-123: ‘We find that, while both precipitation and temperature are robustly projected to increase in the Pacific’ Add ‘tropical Pacific’?

This has been changed.

I realize that the authors define the “Pacific” as only reaching from -5 to 5 N in the methods section but this is not a generally agreed definition of the Pacific so I would strongly suggest to be clear in the language here in order to not confuse the reader.

Pacific has been changed to read tropical Pacific in all cases.

L123-124: ‘Although the SMILE mean shows an increasing gradient’ Are the authors referring to the black line between west and east Pacific in figure 5a here? I’m not able to detect any slope in this line.

This has been changed to read ‘slight increasing gradient’ on line 306.

L135-136: ‘The change in precipitation variability in the far western Pacific varies widely among models’ This does not seem to reflect figure 5d, where in fact the model spread in the Far West Pacific is smallest?

We have changed this on lines 321-322 to reflect that there are limited changes in this variable in the far west Pacific.

L144: “non-independent groups” This seems a bit ambiguous to me. The models with in a group are “non-independent” but the groups are then considered independent, correct?

This line no longer exists in the revised paper. We now refer to the CMIP5 atmospheric subsets as ‘sub-ensembles’ in the text.

L157: I have to admit I’m a bit surprised by this. Why are the authors using a 1 degree grid? I would assume that this means effectively down-scaling most of the models, correct?

See here for grid ranges: <https://portal.enes.org/data/enes-model-data/cmip5/resolution>

The atmospheric models have a range of about 0.75 – 4 degrees, the ocean 0.3-1.5 degrees.

This effectively downscales temperature over land, but upscales over the ocean. As such this choice seems reasonable.

L203: ‘Nature’

This has been fixed.

Reviewer #3 (Remarks to the Author):

Review of Maher et al. 2019

In this paper, the authors present a method to evaluate robustly model-to-model agreement in climate projections based on Single Model Initial-conditions Large Ensembles (SMILEs) and describe its application to some specific cases. The method in itself is not new, but it was not possible to use it before in practice because ensembles of SMILEs did not exist. The authors also describe an alternative new method that can be used when only one or a few members are available for each climate model, based on the fact the climate models are often not independent. They use simulations from “similar” models as members of the same model. This method has some limitations that should better be acknowledged.

We thank the reviewer for their time. This reasons for the differences between the SMILE and the CMIP5 methodology are now discussed in detail on lines 200-226.

The paper is clear, the results are sound and interesting. They are however not especially “cutting-edge” and I wonder whether it is justified to publish this work in a high-impact journal, but it is more a question for the editorial team. It is true that it is the first or one of the first paper to use a significant number of SMILEs to assess model-to-model agreement, but I also have the feeling that they could apply this approach to more interesting cases. And the number of SMILEs used (6) is still very small: it is necessary to remain cautious with the results shown. That being said, I think the paper needs moderate revisions before being publishable.

We understand the limitations of only having 6 SMILEs, however there have been many previous studies only using one SMILE and having 6 is really a large increase on the previous work that has been done.

We feel this work is cutting-edge and important for the following reasons:

- 1. This is the first study to partition the changes in temporal temperature and precipitation variability into model-to-model differences and internal variability*
- 2. The partitioning method for CMIP5 of grouping models that share a component will be extremely useful given the new CMIP6 archive, which has a large disparity in ensemble members available for each model.*
- 3. Understanding the source of uncertainty in projections is important not just for climate modelers, but those trying to understand the impacts of projected changes (see implications on lines 369-373).*

We have additionally added Figure 9 to describe what the methods in this paper add to the traditional multi-model ensemble approaches.

Major comments.

-The focus of the paper is not always clear. At first, we have the impression that the paper is mainly methodological but then, an important part is dedicated to some applications. But the authors don't go very “deep” in these applications.

It is not really a problem, but for example I wonder whether the introduction is totally adapted to the actual content of the paper.

The introduction has been substantially revised.

-A better discussion about the existing ways to characterize model agreement is needed, I think, as several methods have been proposed (e.g. the discussion in the IPCC AR5). For example, it is possible to characterize “robustly” model agreement with a single member per model and piControl

simulations: we can characterize whether the models agree on a significant (greater than internal variability) change. The approach of the authors allows to show model agreement even if the future change is smaller than internal variability for individual models. This should be discussed, I think.

Discussion has been added on lines 60-68.

Also, the question of model agreement basically lies on the one of the estimation of the forced signal and different methods can be used in that context. It is possible to do the same analyses as in this paper but with a different method to estimate the forced signal (e.g. spline, loess-filter etc. for example). Some authors have used this approach (e.g. Hawkins and Sutton 2009) to separate internal variability and the forced signal with a single member per model. I personally think that this approach has major limitations, and that using SMILEs to estimate the signal is the only method that is really satisfactory.

But I think that this point should be discussed, in order for the authors to convince all readers of the interest of their approach.

We mention the limitations of previous methods on lines 45-50. Repeating the analysis with multiple methods that we already know are limited is out of the scope of the paper. The fact that the SMILE and CMIP methods agree well supports the choice of the methods used in our study.

-The study area looks strange to me. Why do the authors stop their maps at 40° N and S? It would be possible to show global maps or at least maps with mid-latitudes. It is all the stranger because the authors discuss the results for Europe, but only the extreme south of Europe is visible on the maps. Spatial averages over “Europe” are shown but Europe is defined as 50°E – 150° E and 20° N – 40°N. None of these latitudes / longitudes makes sense to define Europe. With this definition, India is at the center of Europe ! Maybe the authors meant -50°E but even in that case the latitudes do not make sense. Cairo is not in Europe. Such weird definition of regions is also seen for North-America, whose definition excludes a large part of USA, and Canada in totality. This is really misleading.

I have the feeling (based on what is visible for the northern part of the domain in Figure 1e) that the authors may use their peculiar study area because their approach based on the re-sampling of CMIP5 models does not work at higher latitudes. Is that the case? Why not simply say it (and show it)?

We have extended our Figures to include the entire globe and have redefined the regions we discuss over land based on new IPCC regions defined by Iturbide et al.

-I think the authors overestimate the interest of their approach to estimate model-to-model agreement based on the re-sampling of single (or few) members from CMIP5 models that share a common atmospheric / oceanic model. It is true that two models that share some common components should not be considered independent, but it is also too extreme to consider that two coupled climate models that share the same atmospheric model are in fact the same model and that their differences are only due to internal variability. And it is clearly visible in the results of some analyses of the paper. The role of “internal variability” is often largely greater in Figure 1e than in 1b (even if it is impressive to see the differences between Figure 1e and 1f). This should be acknowledged.

I don't suggest to remove these analyses, which are interesting, but I think the discussion needs to be more balanced, and the limitations need to be discussed: the method can give an indication but it is far from perfect etc...

I also think it would be interesting to better illustrate how the lack of independence impacts model agreement, e.g. compare CMIP5 model agreement with their approach and the classical approach (based on the hypothesis that all models are independent).

We now discuss the differences between SMILE analysis and CMIP analysis in detail on lines 200-226 and have added Supplementary Figure S10 to address these differences.

We feel this method is extremely interesting especially given the upcoming CMIP6 database where this method could be useful and applied, we stress this on lines 225-226 and 345-348.

We cannot compare the CMIP5 sub-ensemble result to a result where we assume all CMIP5 models are independent for the partitioning of uncertainty as in this case we cannot separate internal variability and model differences due to the lack of ensemble members for most CMIP5 models. The forced changes themselves (left panels Figures 1-4) compared well with previous work as mentioned in the text. We have, however, now added Figure 9 to compare CMIP5, the atmosphere sub-sampled CMIP5 and the SMILE results in three example regions. We discuss on lines 358-373 the added value of our analysis to previous methods.

Minor comments

L16-17 “largely independent”: please describe the models somewhere, with the components etc. How GFDL-CM3 and GFDL-ESM2M can be said to be independent? They have a lot in common, haven't they?

We describe this on lines 127-136.

L19-21. Not clear. I had to read the whole manuscript to understand what is meant here.

This has been reworded on lines 17-18.

L26. What has been done wrong in 7 and 8 ?

This line is no longer in the manuscript

L51-52. OK. But the reference 15 only deals with precipitation biases over the South Asian. It is not sufficient to substantiate the affirmation (however, it is true that looking at Figure 5, GFDL-ESM2M and GFDL-CM3 do seem to behave differently).

This is correct. A new study by Lehner et al 2020 has looked in more detail at the differences between the two models. We have added this to lines 127-136.

L62. Please refer to the method section here and where necessary. At first, I wondered how some analyses were done, whereas they are explained in the Method section.

We have restructured the manuscript to better describe the methods at the beginning of the results section on lines 94 onwards.

L65-67. In the Tropical Pacific, the internal variability is not especially large in Figure 1b,e. Do you talk about precipitation or temperature variability change? Please be more precise.

This section has been revised to be more precise.

L66. “even if the models agree on projected change”. Based on which analysis? Or on which

references? Do you refer to your analysis in Figure 5? In that case, say it, please. But as shown in Figure 5, models do not agree on precipitation variability change. So you have to be more precise.

This was meant to be a general statement. We have revised the text to start with this general statement, then include precise examples. See lines 152-157.

L77. “Accurately” is quite exaggerated to my opinion. It is interesting and it is true that it is quite spectacular to see how models from the same pool behave similarly compared to the other models (Figure 1e versus 1f). But we can still see that the “internal variability change” with this approach (Figure 1e) is largely greater than the one estimated with SMILEs, maybe two times greater, even if the color palette does not make this estimation easy.

We no longer use the word accurately. The differences are discussed in more detail on lines 200-226.

L89 and Figure 3. You could use two symbols in Figure 3, to show where the CMIP5 models agree on an increase and where they agree on a decrease. Is there at some places an agreement in the SMILEs on an increase and an agreement in the CMIP5 models on a decrease?

This has been done.

L94. See one of my major comment.

We have revised this section to be more precise.

L119-120. It would be interesting to further discuss the differences between the results of 23 and 24. Is it the way the models are analyzed? Is it because the two studies use different samples of models? We can guess from lines 121 to 127 but a sentence would be nice.

We have added that this occurs in one model, which is the difference.

L141-142. “For the first time...”. Arguably Hawkins and Sutton (2009) already did that. I would say that it is done for the first time “correctly”, without to have to use an approximate estimate of the forced signals.

We now use first time only to refer to temporal variability, where this had not been done before.

L150. Agreed. You could add a word of caution about the limitations to the results in this paper due to the small number of SMILEs available today.

This is now in the supplementary discussion. We also discuss the SMILEs in more details on line 111-136.

L160-161. Given the approach proposed by the authors to assess agreement with CMIP5 models would it not been more logical and consistent to detrend the CMIP5 models using the ensemble means within each group?

It would be more correct to do this. However, some sub-ensembles using CMIP5 only have one model in them so we cannot detrend in this fashion. We want to keep these groups as they contribute to the model differences, which is why we use the traditional detrending approach.

Reviewer #1 (Remarks to the Author):

Second review of ‘More accurate quantification of model-to-model agreement in externally forced climatic responses over the coming century’ Overview I thank the authors for responding to my first review. In general, I disagree with their argument that it is not necessary to assess model fidelity because ‘model-to-model agreement without considering model fidelity is the usual method of choice to assess uncertainty’, but I also recognize that there are limitations to how much work can go into a single paper. I simply recommend, as per major point 3, that the authors be clear that the results presented indicate the behavior across models, and do not necessarily give us quantitative predictions of the real world.

Similar to Reviewer 3, I remain unsure whether this paper belongs in a high-profile journal given (1) the number of papers over the past five years that have highlighted the role of internal variability on annual mean quantities, (2) the lack of assessment of physical mechanisms associated with any of the more interesting changes that are highlighted, and (3) the lack of validation of any of the model results. That said, the manuscript does provide a number of figures that will be helpful for reference as the SMILEs are evaluated in the future, as well as a first general quantification of the cross-model agreement of forced changes in variability.

Major points 1.

The text and equations from line 98-107 are unclear and incorrect as written. First, based on Fig. 1, it appears that ΔTFS is constant in time, so the authors need to add an indication of time averaging to their equation. Second, equation (2) is incorrect.

Based on Fig. 1, and the points that the authors are making in text, presumably equation (2) is intended to capture the variability in ΔTFS that can occur due to the existence of internal variability. This should thus be written as: $\sigma(\Delta T) = \text{std}(\overline{T}^{\text{future}} - \overline{T}^{\text{past}})$, where the overbars indicate a time average over the respective periods and the standard deviation is being taken across ensemble members in a given ensemble. Equation (3) does not make sense: the ‘change in T’ cannot be calculated by the sum of a best-estimate mean change and the standard deviation of that estimate.

Finally, it seems the uncertainty should also account for differences across the large ensembles in their internal variability, just as it does with their mean. In the corrected equation 2, the spread of internal variability across the SMILEs could be calculated and used instead of the mean across each model. 2. In order to be clear about the method, the authors should also provide equations or clear text indicating how they are calculating the forced changes in temperature and precipitation variance as presented in Figs. 3 and 4. 3.

When the authors discuss confidence (i.e. lines 257-259), what do they mean they 1 have confidence in? Given the lack of independence of the SMILEs, and the lack of validation, I hesitate to say that we have confidence that the real world will follow these model projections.

Minor points:.

Line 43: Suggest ‘making it non-trivial to draw generalizations about which uncertainty dominates’ or similar 2. Equation (1): Use T (rather than t) for temperature for consistency. 3. Line 109: The method of detrending (mentioned in the methods) should be here as well for clarity. 4. Line 125: replace the comma with a semicolon 5. Line 152: this sentence is misleading. Assuming that the uncertainty due to internal variability and model differences are independent, then the uncertainties

add, so a single ensemble will not be effective at spanning the full uncertainty. 6. Tables S3 and S4: please provide the total number of ensemble members for each group.

Reviewer #2 (Remarks to the Author):

Second review of "More accurate quantification of model-to-model agreement in externally forced climatic responses over the coming century" by N. Maher, S. Power, and J. Marotzke.

This is my second review of this manuscript. I acknowledge the large effort the authors seem to have put into revising the manuscript and I now find it to be quite concise. In addition they have included new figures into the main paper and the appendix which provide new insights. I think this work provides valuable insights by diving into the question of how to cleanly separate internal variability and forced response focusing on global maps as well as some selected regions. I've some minor comments otherwise I think this manuscript can be published in Nature Communications.

Minor comments:

L20: add 'latitude' after 80°?

L29: delete 'to increasing greenhouse gases'? The RCPs (and new SSPs in particular) also represent other changes such as aerosols and land use change.

L73: delete 'which'

L87: SMILE has already been introduced?

L89: Maybe don't call it 'disagreement'?

L116/Figure S1: Could the authors adjust the colorbar for the majority of the plots or maybe plot show the difference relative to the absolute change of the 100 member ensemble. It seems to me that for the 20 member samples (which are somewhat most relevant as the authors try to argue for using GFDL-CM3) the difference is actually around 20% at high latitudes but it is really hard to tell.

Figure3/4: I think it might be worth to consider changing the colourmaps used in the three leftmost columns of these figures. Taking the example of Figure 4a: this shows the change in precipitation variability, yet it uses the same colourmap as Figure 2a. The brown/green colour scheme also visually indicates decreases/increases in precipitation which I find to be misleading.

L304: Figure 8a, c

L319: Figure 8b, d

Reviewer #3 (Remarks to the Author):

Major modifications have been made to the paper following the three previous reviews, and the paper has much improved. However, some important issues remain (some of them already mentioned in the previous reviews). Some definitions and equations do not make sense. Notations are imprecise. Major limitations of the methodology are not discussed. The paper is therefore not ready for publication and major revisions are needed.

L98. I don't see why it is called "forced change". Usually, it is simply called "change"...

L99. t is the climatological mean of temperature and not the temperature, right? Please be more rigorous with the definitions and notations.

L102. "The internal variability in T ". I disagree: based on the formula it is (an estimation of) internal variability in ΔT , not T .

I suppose that std means "standard deviation"? And it is better to be consistent: please give the formula as you did for equation 1 (you didn't write $\Delta T = \text{mean}(t_{\text{future}} - t_{\text{present}})$) Also, the sentence is not complete, there is no verb.

L103. This equation does not make sense. "The projected change in T " is simply given by equation (1). Adding the standard deviation to the mean change and calling that "projected changes" is not meaningful.

L104. Please write a complete sentence (no verb).

L104. I also disagree with this equation. We know how to compute the real uncertainty (in terms of standard deviation) in the projected change: it is the weighted standard deviation of all the SMILEs members, the weights being given by the inverse of the ensemble sizes. It is clearly not equal to the sum of the two terms in equation 4, standard deviations are not additive (By the way, this point had already been made by a reviewer in the first reviews.)

L109: As I said in my previous review, please refer to the method section where necessary. I wondered how the quantities were detrended and didn't know it was explained in the Method section.... Note that the word detrended alone is somewhat misleading for the operation done here. The term "demeaned" is quite commonly used nowadays for this operation.

L202 "same family": in practice it is even "same model", right?
Please refer to the method section.

L215 etc. Please refer to the figure number. It is not obvious to guess that we have to go 4 figures back and return to Figure 1...

L217-218 "due to the overestimation on internal variability". It is necessary to explain why, to discuss the limitations of the methods used in a paper.

This is explained by the strong approximations of the method used for the CMIP5 models. With this method a (probably quite large) part of "internal variability" is actually due to inter-model differences. Even if the results of climate models with the same atmosphere model tend to be close (closer, at least), they are not identical and therefore their difference does not characterize internal variability only.

L225. Regarding the "reasonable estimate of U_{iv} ". Please show the ratio of U_{iv} between Figure 1f and 1b. I'm pretty sure that there is a factor of at least 2 in many regions and sometimes maybe even 10 between the two estimates of U_{iv} . Given such a strong overestimation for A-CMIP5, I think it is very difficult to talk about a "reasonable estimate". This should be discussed (see the previous point).

Figure 9: What is the "bootstrapped error" on the mean? How is it computed exactly?

REVIEWER COMMENTS

We thank all reviewers for their helpful comments, which we believe have significantly improved the manuscript. We wish to point out the following two points:

1. Based on the review comments we have identified issues with equations as written in the text and have updated them. It was additionally pointed out that it is not correct to average standard deviations. For this reason, we now use variance in many of the calculations as shown in equations 5, 6, 7, 8, 9, 11 and 12 as well as use variance to aggregate spatially. This methodology is discussed in detail in the methods on lines 468-477 and 493-494 as well as in the equations and their description on lines 98-140.

2. We found a small error in the code, that only affects the mean-state temperature projections in ARO (panel a in Figure 6). This increases the model differences in the Arctic as compared with the previous version of the manuscript, which agrees well with other studies: see lines 408-411

Reviewer #1 (Remarks to the Author):

Second review of ‘More accurate quantification of model-to-model agreement in externally forced climatic responses over the coming century’

We thank the reviewer for taking the time to complete this second in-depth review and have revised the manuscript substantially in response to the comments provided below.

Overview

I thank the authors for responding to my first review. In general, I disagree with their argument that it is not necessary to assess model fidelity because ‘model-to-model agreement without considering model fidelity is the usual method of choice to assess uncertainty’, but I also recognize that there are limitations to how much work can go into a single paper. I simply recommend, as per major point 3, that the authors be clear that the results presented indicate the behavior across models, and do not necessarily give us quantitative predictions of the real world.

We acknowledge this limitation and have added the following sentences to address it in the discussion section (lines 424-428): ‘In this paper we have focused on the degree of model-to-model agreement. This is an important issue to consider for assessing the degree of confidence we have in a projected change. Other important factors to consider include the degree to which models simulate related observed properties, the plausibility of the mechanisms responsible for the change, and the statistical consistency between the observed trend and the projected change (see Power et al. [e.g. 1]).’

Similar to Reviewer 3, I remain unsure whether this paper belongs in a high-profile journal given (1) the number of papers over the past five years that have highlighted the role of internal variability on annual mean quantities, (2) the lack of assessment of physical mechanisms associated with any of the more interesting changes that are highlighted, and (3) the lack of validation of any of the model results. That said, the manuscript does provide a number of figures that will be helpful for reference as the SMILEs are evaluated in the future, as well as a first general quantification of the cross-model agreement of forced changes in variability.

We argue that this manuscript is suited for this journal for the following reasons:

- 1) *This will be the first paper to investigate sources of uncertainties in variability itself and to provide a new method for separating uncertainties in multi-model ensembles, which is why we argue that it is appropriate for a high-profile journal*
- 2) *While we acknowledge the importance of investigating physical mechanisms, assessing such mechanisms for each topic in the paper would be the the amount of work of a single paper each and is out of the scope of our study. We mention on line 298 that these examples are meant “to illustrate how the SMILE results can be used”.*
- 3) *We have added text around the limitation of not assessing model fidelity on lines 424-428*

Major points

1. The text and equations from line 98-107 are unclear and incorrect as written. First, based on Fig. 1, it appears that ΔTFS is constant in time, so the authors need to add an indication of time averaging to their equation. Second, equation (2) is incorrect. Based on Fig. 1, and the points that the authors are making in text, presumably equation (2) is intended to capture the variability in ΔTFS that can occur due to the existence of internal variability. This should thus be written as: $\sigma(\Delta T) = \text{std}(\overline{T^{\text{future}}} - \overline{T^{\text{past}}})$, where the overbars indicate a time average over the respective periods and the standard deviation is being taken across ensemble members in a given ensemble. Equation (3) does not make sense: the ‘change in T’ cannot be calculated by the sum of a best-estimate mean change and the standard deviation of that estimate. Finally, it seems the uncertainty should also account for differences across the large ensembles in their internal variability, just as it does with their mean.

Thanks for pointing this out. We have revisited and completely revised this section based on the review comments. Specifically we have updated the equations to include the overbars to indicate the time averaging. We have removed Equation 3 and instead replaced it with a new equation 1, which gives us the projected change in each individual ensemble member. We have also updated the internal variability equation to use the sigma symbol.

Rather than claim that we address all uncertainties, we now clarify that we quantify the uncertainty due to model differences in the forced change (line 102-103), and internal variability (line 103). We no longer claim that the addition of these two terms covers all uncertainty.

See the Results section beginning on line 98 for all changes in the equations.

In the corrected equation 2, the spread of internal variability across the SMILEs could be calculated and used instead of the mean across each model.

The equations have been changed to be in line with Rowell et al (1995). This includes a corrected equation for internal variability across the SMILEs in equation 5.

2. In order to be clear about the method, the authors should also provide equations or clear text indicating how they are calculating the forced changes in temperature and precipitation variance as presented in Figs. 3 and 4.

We have updated the text on lines 128-140 to address this point and added equations 10-12 for the forced changes in temperature and precipitation temporal variability.

3. When the authors discuss confidence (i.e. lines 257-259), what do they mean they have confidence in? Given the lack of independence of the SMILEs, and the lack of validation, I hesitate to say that we have confidence that the real world will follow these model projections.

We have revised the text from lines 272-298 based on this comment. We now discuss model-to-model agreement rather than confidence in the text.

Minor points

1. Line 43: Suggest ‘making it non-trivial to draw generalizations about which uncertainty dominates’ or similar

This has been changed as suggested (see line 48-49)

2. Equation (1): Use T (rather than t) for temperature for consistency.

This has been changed as suggested.

3. Line 109: The method of detrending (mentioned in the methods) should be here as well for clarity.

This has been done, we have added ‘To compute T_{var} first we remove the forced response by removing the ensemble mean at each timestep’ on lines 128-129.

4. Line 125: replace the comma with a semicolon

This has been done.

5. Line 152: this sentence is misleading. Assuming that the uncertainty due to internal variability and model differences are independent, then the uncertainties add, so a single ensemble will not be effective at spanning the full uncertainty.

We now say ‘combined variance’ on line 184 we no longer claim that we address all uncertainties or that we can simply add the two types of uncertainty,

We have changed the sentence (previously on line 152) to read (line) “Where U_{IV} is of a similar magnitude to U_{MD} , an individual SMILE could cover the uncertainty in the forced response.” (now lines 185-186)

6. Tables S3 and S4: please provide the total number of ensemble members for each group

This has been done.

Reviewer #2 (Remarks to the Author):

Second review of “More accurate quantification of model-to-model agreement in externally forced climatic responses over the coming century” by N. Maher, S. Power, and J. Marotzke.

This is my second review of this manuscript. I acknowledge the large effort the authors seem to have put into revising the manuscript and I now find it to be quite concise. In addition they have included new figures into the main paper and the appendix which provide new insights. I think this work provides valuable insights by diving into the question of how to cleanly separate internal variability and forced response focusing on global maps as well as some selected regions. I’ve some minor comments otherwise I think this manuscript can be published in Nature Communications.

We thank the reviewer for their positive feedback and the time taken to review this manuscript.

Minor comments:

l20: add 'latitude' after 80°?

This has been added.

l29: delete 'to increasing greenhouse gases'? The RCPs (and new SSPs in particular) also represent other changes such as aerosols and land use change.

This has been changed to 'external forcing' on line 29

L73: delete 'which'

Deleted.

l87: SMILE has already been introduced?

We have deleted the full text name and left the acronym.

L89: Maybe don't call it 'disagreement'?

We have changed the word to differences

L116/Figure S1: Could the authors adjust the colorbar for the majority of the plots or maybe plot show the difference relative to the absolute change of the 100 member ensemble. It seems to me that for the 20 member samples (which are somewhat most relevant as the authors try to argue for using GFDL-CM3) the difference is actually around 20% at high latitudes but it is really hard to tell.

We have changed the differences to % differences for Figures S1-S4 and updated the text which explain these Figures to match.

Figure3/4: I think it might be worth to consider changing the colourmaps used in the three leftmost columns of these figures. Taking the example of Figure 4a: this shows the change in precipitation variability, yet it uses the same colourmap as Figure 2a. The brown/green colour scheme also visually indicates decreases/increases in precipitation which I find to be misleading.

Intuitively we think it makes sense to plot temperature and temperature variability using the same colourmap and precipitation and precipitation variability on the same colourmap. As such we have chosen to keep the colourbars as they are.

L304: Figure 8a, c

Fixed.

L319: Figure 8b, d

Fixed.

Reviewer #3 (Remarks to the Author):

Major modifications have been made to the paper following the three previous reviews, and the paper has much improved. However, some important issues remain (some of them already mentioned in the previous reviews). Some definitions and equations do not make sense. Notations are imprecise. Major limitations of the methodology are not discussed. The paper is therefore not ready for publication and major revisions are needed.

We thank the reviewer for their time. The review comments below have been addressed and have improved the manuscript.

L98. I don't see why it is called "forced change". Usually, it is simply called "change" ...

This is forced change because we average over the ensemble to remove internal variability and estimate the change due to external forcing alone. We have added the following text on lines 100-102 to clarify this: "We calculate the projected change in an individual ensemble member; the forced change both in each individual SMILE and across the six SMILEs, which is an estimate of the projected change due to external forcing alone"

L99. \bar{t} is the climatological mean of temperature and not the temperature, right? Please be more rigorous with the definitions and notations.

We have updated the symbol and included an overbar see lines 106-107 and equation 1

L102. "The internal variability in T ". I disagree: based on the formula it is (an estimation of) internal variability in ΔT , not T .

I suppose that std means "standard deviation"? And it is better to be consistent: please give the formula as you did for equation 1 (you didn't write $\Delta T = \text{mean}(t_{\text{future}} - t_{\text{present}})$) Also, the sentence is not complete, there is no verb.

You are correct. We have updated this to be ΔT not T on line 115

We have additionally updated std to σ for consistency as suggested (see equation 4)

We have also added a verb to the sentence.

L103. This equation does not make sense. "The projected change in T " is simply given by equation (1). Adding the standard deviation to the mean change and calling that "projected changes" is not meaningful.

Apologies, we agree. We have removed this equation and added a new equation (now eq 1) to demonstrate the projected change in a single ensemble member.

L104. Please write a complete sentence (no verb).

This has been fixed.

L104. I also disagree with this equation. We know how to compute the real uncertainty (in terms of standard deviation) in the projected change: it is the weighted standard deviation of all the SMILEs members, the weights being given by the inverse of the ensemble sizes. It is clearly not equal to the

sum of the two terms in equation 4, standard deviations are not additive (By the way, this point had already been made by a reviewer in the first reviews.)

We have removed this equation and no longer claim to assess total uncertainty. We investigate each uncertainty individually. In the captions for Figures 1-4 we note that we look at the percentage variance contribution of U_{MD} to the sum of U_{MD} and U_{IV} not the total uncertainty.

We apologise as we missed the full implications of the non-additivity of standard deviations in the first round of reviews. We now average variance as shown in equations 5, 6, 7, 8, 11 and 12 as well as use variance to aggregate spatially. This methodology is discussed in detail in the methods on lines 468-477 and 493-494 as well as in the equations and their description on lines 98-140.

L109: As I said in my previous review, please refer to the method section where necessary. I wondered how the quantities were detrended and didn't know it was explained in the Method section.... Note that the word detrended alone is somewhat misleading for the operation done here. The term "demeaned" is quite commonly used nowadays for this operation.

We have added the sentence "To compute $Tvar$ first we remove the forced response by removing the ensemble mean at each timestep." to address this on lines 128-129.

L202 "same family": in practice it is even "same model", right?
Please refer to the method section.

We have clarified this on lines 238-240 which reads "We note that the sub-ensembles include both different models, which share an atmospheric component and multiple ensemble members from the same model where available."

We have also referred to the methods on line 237.

L215 etc. Please refer to the figure number. It is not obvious to guess that we have to go 4 figures back and return to Figure 1...

This has been added.

L217-218 "due to the overestimation on internal variability". It is necessary to explain why, to discuss the limitations of the methods used in a paper.

This is explained by the strong approximations of the method used for the CMIP5 models. With this method a (probably quite large) part of "internal variability" is actually due to inter-model differences. Even if the results of climate models with the same atmosphere model tend to be close (closer, at least), they are not identical and therefore their difference does not characterize internal variability only.

We explain this on lines 245-247 . These lines have been expanded from the previous version to clarify the point to read: "First, U_{IV} could be overestimated in the CMIP5 analysis as the models in the sub-ensembles are not the same. This means that the uncertainty estimated as U_{IV} contains some of the uncertainty from U_{MD} ."

L225. Regarding the "reasonable estimate of U_{iv} ". Please show the ratio of U_{iv} between Figure 1f and 1b. I'm pretty sure that there is a factor of at least 2 in many regions and sometimes maybe even 10 between the two estimates of U_{iv} . Given such a strong overestimation for A-CMIP5, I think it is

very difficult to talk about a "reasonable estimate". This should be discussed (see the previous point).

We now include Supplementary Figure 11, which has this information. We discuss the estimation of U_{IV} in more detail in the paragraph beginning on line 251 and have removed the word reasonable, to let the reader decide if they think it is reasonable based on the information provided. We acknowledge the large differences between the methods for temperature on lines 255-256 "We find that the estimate of U_{IV} for ΔT is much larger using CMIP5 than the SMILEs (Figure S11a)"

Figure 9: What is the "bootstrapped error" on the mean? How is it computed exactly?

This has been clarified for Figures 6-9 in the captions. Errorbars are computed by bootstrapping 1000 times with the matlab bootci function for the mean. For Figure 9 we have clarified in the caption: with the bootstrapped error on the mean shown in the small black errorbars (1000 samples using matlab bootci).

Reviewer Comments, third round –

Reviewer #1 (Remarks to the Author):

I thank the authors for responding to my prior review. The revised draft is improved, and may be suitable for publication. I have a number of minor comments, listed below. I also suggest that the authors make their code available online in e.g. a github repository, and mention the availability in the 'Data Availability' section.

1. Line 115: Recommend starting the sentence with 'The spread across the ensemble due to internal variability...' since the equation is not really summarizing internal variability itself.
2. Line 117: Change 'The uncertainty...' to 'An estimate of the uncertainty...'
3. Line 129: I found the notation of $Tvar$ confusing, since it makes me think of variance, but is actually a standard deviation. Perhaps σ ? It tripped me up in Eqn. 11 because I thought the variance was being additionally squared.
4. Line 132: For variance calculations, it is necessary to say what temporal resolution is considered in the variance calculation.
5. Line 135: Unclear what this line is saying.
6. Line 185: 'Where U_{IV} is of a similar magnitude to U_{MD} ...' should read 'Where U_{IV} is much greater than U_{MD} ...' for the following clause to be true (since the two types of uncertainty add together).
7. Line 424: Unclear what this sentence is saying. Are the authors indicating that simply choosing one member of each CMIP5 model provides the same information as their sub-ensemble approach?

Reviewer #3 (Remarks to the Author):

The authors have made important modifications to the paper and dealt with the issues I noted in my previous review. I think the paper is now ready for publication, from a scientific standpoint. Note that because of the very extensive changes done in response to the reviews since the submission, the paper seems rather long for a letter now, and its main point is not very clear, as several (interesting) questions are now dealt with in the paper, at the same level.

Reviewer #1 (Remarks to the Author):

I thank the authors for responding to my prior review. The revised draft is improved, and may be suitable for publication. I have a number of minor comments, listed below. I also suggest that the authors make their code available online in e.g. a github repository, and mention the availability in the 'Data Availability' section.

We thank the reviewer for taking the time to complete a third review. The Max Planck Institute for Meteorology has it's own repository to make code available. The code will be made available in this location: <http://hdl.handle.net/21.11116/0000-0007-4AFD-A>

1. Line 115: Recommend starting the sentence with 'The spread across the ensemble due to internal variability...' since the equation is not really summarizing internal variability itself.

This has been updated as suggested on line 109

2. Line 117: Change 'The uncertainty...' to 'An estimate of the uncertainty...'

We have changed this. See line 109-110

3. Line 129: I found the notation of T_{var} confusing, since it makes me think of variance, but is actually a standard deviation. Perhaps T_{σ} ? It tripped me up in Eqn. 11 because I thought the variance was being additionally squared.

Thank you, this is really helpful and we have changed it as suggested throughout the text and in the Figures.

4. Line 132: For variance calculations, it is necessary to say what temporal resolution is considered in the variance calculation.

We have added this information on line 110. It reads: "In this study we will investigate the externally forced response of annual-mean temperature (T), annual-mean precipitation (P), annual-mean temporal temperature variability (T_{σ}) and annual-mean temporal precipitation variability (P_{σ})"

5. Line 135: Unclear what this line is saying.

We have modified the line (now lines 117-118) for clarity to read "here the standard deviation is calculated individually for each time period as the square root of the ensemble mean variance before the difference between the two time periods is calculated."

6. Line 185: 'Where U_{IV} is of a similar magnitude to U_{MD} ...' should read 'Where U_{IV} is much greater than U_{MD} ...' for the following clause to be true (since the two types of uncertainty add together).

We do not mean to say that U_{IV} could cover the total uncertainty, but that it could cover U_{MD} itself. We have updated this for clarity on lines 167-168 to read "Where U_{IV} is of a similar magnitude to U_{MD} , an individual SMILE could cover the uncertainty in U_{MD} itself"

7. Line 424: Unclear what this sentence is saying. Are the authors indicating that

simply choosing one member of each CMIP5 model provides the same information as their sub-ensemble approach?

Yes for this specific example this is what we mean. We have, however, removed this sentence to improve the flow of the discussion and now refer to this result on line 416

Reviewer #3 (Remarks to the Author):

The authors have made important modifications to the paper and dealt with the issues I noted in my previous review. I think the paper is now ready for publication, from a scientific standpoint. Note that because of the very extensive changes done in response to the reviews since the submission, the paper seems rather long for a letter now, and its main point is not very clear, as several (interesting) questions are now dealt with in the paper, at the same level.

We thank the reviewer for taking the time to look over this paper again. We are happy to hear that you think the paper is now ready for publication.